# Cohort-resolved excess mortality in Germany (2000-2024): Patterns and implications for the SARS-CoV-2 era

**Robert Rockenfeller**[1]*, **Michael Günther**[2,3]

**1** Mathematical Institute, University of Koblenz, Koblenz, Germany, **2** Computational Biophysics and Biorobotics, Institute for Modelling and Simulation of Biomechanical Systems, University of Stuttgart, Stuttgart, Germany, **3** Friedrich–Schiller–Universität, Jena, Germany

* rrockenfeller@uni-koblenz.de

**Data availability statement:** All relevant data are within the manuscript and its Supporting information files.

## Abstract

Understanding the impact of the SARS-CoV-2 pandemic on mortality requires more than aggregate statistics. While whole-population indicators have informed policy, they risk concealing subgroup-specific patterns. We analysed all-cause mortality in Germany from 2000 to 2024 using a weekly, cohort-resolved framework across 15 age groups to detect excess and under-mortality before, during, and after the pandemic.

Expected mortality was modelled using exponential trends from two decades of pre-pandemic data. Deviations from expectation were quantified as normalised excess all-cause mortality rates (NEAMR), enabling the identification of significant, age-specific anomalies.

We found sustained NEAMR in adults aged 75-79 and 35-49 from late 2021 through 2024—patterns absent in whole-population trends. Conversely, cohorts aged 30-34 and 55-59 showed persistent under-mortality. Earlier excess peaks in older cohorts (e.g., 85-89 in 2003, 95+ in 2013) suggest generational vulnerabilities potentially linked to early-life adversity. Cross-correlation analyses indicate that associations between NEAMR and SARS-CoV-2 mRNA injection rates diverge from expected protective patterns in most age cohorts, especially during the 2021 'alpha-to-delta' transition. These findings highlight the need for further hypothesis-driven investigations as well as a high-resolution mortality surveillance. Cohort-resolved analysis reveals NEAMR signals that aggregate data obscure, offering a more accurate assessment of public health outcomes across demographic groups.

## 1 Introduction

Until 2009, the global public health community broadly adhered to a consensus definition of pandemics. Namely, they involved the emergence of a novel pathogen to which there was no human immunity, and which caused widespread illness and substantial mortality. The World Health Organization (WHO), for instance, described an influenza pandemic as occurring "when a new influenza virus appears against which the human population has no immunity, resulting in epidemics worldwide with enormous numbers of deaths and illness" [1]. Central

**Funding:** The author(s) received no specific funding for this work.

**Competing interests:** The authors have declared that no competing interests exist.

to such assessments was the empirical evaluation of (excess) all-cause mortality (AM). This indicator is a robust and objective measure of population-level health crises.

After 2009, however, the definitional framework employed by the WHO underwent notable revisions [2]. These changes effectively decoupled the pandemic label from quantitative criteria such as morbidity and mortality thresholds and allowed the declaration of a pandemic in March 2020 [3]. In the case of SARS-CoV-2, the decision rested on grounds that no longer explicitly required demonstrable excess mortality (EM) or widespread health system failure.

In Germany, this declaration had immediate and far-reaching implications. Following the WHO's announcement on March 11th, 2020, the German Bundestag, on March 25th, 2020, enacted emergency legislation based on an "epidemic situation of national importance" ("epidemische Lage von nationaler Tragweite"). This legislation granted the federal and local governments powers to restrict fundamental civil liberties – for example the implementation of extended lockdowns, mask mandates, and a nationwide SARS-CoV-2 mRNA injection (mRNA-I) campaign. These interventions, occurring in defined phases, provide a temporal framework that is relevant when analyzing mortality data at weekly resolution.

Because the pandemic was declared worldwide, it is important to check whether the evidence supporting that decision holds up against historical standards. A key standard is excess all-cause mortality (EAM) – the number of deaths above what would normally be expected in a given period. Earlier studies for Germany [4,5] applied the pre-2009 WHO pandemic criteria to an EAM framework and found no sign of unusually high mortality in 2020. Similarly, there was no major increase in deaths during the 2019/2020 flu season (including early 2020) [4, Fig 1], nor during the 2020/2021 season, even after the mass rollout of mRNA-I beginning on December 27th, 2020. That 2020/2021 season recorded about 27,600 excess deaths [4, Table 2], which is similar to the 2017/2018 flu season (about 25,100 excess deaths [6, p. 47]). By contrast, the 2018/2019 and 2019/2020 flu seasons both showed undermortality – roughly 22,000 fewer deaths than expected [4, Table 2].

The 2021/2022 flu season, in contrast, was marked by an EAM of approximately 31,100 individuals [4, Table 2]. This occurred in the aftermath of a sustained mRNA-I campaign and multiple lockdowns, including widespread societal, psychological, and physiological consequences, mandatory masking and healthcare delays. The arrival and predominance of the SARS-CoV-2 'omicron' variant (B.1.1.529) in late 2021 – widely considered milder – complicates causal attribution. The apparent mortality burden of this flu season appears to involve factors beyond viral characteristics, cf. [7] for a detailed state-level analysis in Germany.

This study extends our previous work [4] by examining age-stratified, weekly EAM data in Germany from 2020 through 2024. Our goal is twofold. First, we investigate whether pandemic-level mortality signals may have been masked at the population level but remain detectable within specific age cohorts. Second, we assess whether age-resolved EAM correlates temporally with other epidemiological variables, in particular the weekly, normalised incidence of PCR-positive tests for SARS-CoV-2 (short: PCR-incidence) [8], and mRNA-I rates during the global injection campaign.

## 2 Methods

### 2.1 Abbreviations and mathematical symbols

As this section introduces and handles a variety of concepts, Table 1 offers an overview of all essential abbreviations as well as their respective mathematical symbols used throughout this work.

**Table 1. Abbreviations and mathematical symbols used throughout the study.**

| abbreviation {mathematical symbol} | meaning | description |
|---|---|---|
| SARS-CoV-2 | Severe Acute Respiratory Syndrome Corona Virus 2 | term used consistently for the virus; originally (and more appropriately) named 2019-nCoV |
| SP | spike protein | the spike protein of the SARS-CoV-2 shell |
| mRNA-I | SARS-CoV-2-mRNA injection | injection of any dose of lipid-particle-coated modified messenger RNA that codes for the SP |
| $\{N_{coh}\}$ or $\{N_{pop}\}$ | cohort or population size | number of individuals in a given age cohort or the whole population, respectively |
| $\{t\}$ | time | represented in calendar weeks since the year 2000 |
| AM | all-cause mortality | deaths from any cause |
| AMC $\{D_{AM}\}$ | all-cause mortality count | number of deaths recorded within a given interval (week, year, season) |
| $\{D_{AM,coh}\}$ | cohort-specific mortality count | AMC for a specific age cohort |
| AMR $\{r_{AM,coh}\}$ | all-cause mortality rate | ratio of AMC to the size of the corresponding cohort |
| exAMR $\{r^{ex}_{AM,coh}(t)\}$ | expected AMR | model estimate for $r_{AM}$ at a certain time $t$ |
| NAMR | normalised AMR | AMR normalised to its value in the reference year (2000) |
| exNAMR $\{\tilde{r}^{ex}_{AM,coh}(t)\}$ | expected NAMR | model-predicted NAMR for a given cohort and time |
| NEAMR | normalised excess AMR | relative deviation of AMR from expected values: $(r_{AM,coh} - r^{ex}_{AM,coh})/r^{ex}_{AM,coh}$ |
| EAMC | excess all-cause mortality count | observed AMC minus expected AMC (exAMC) |
| RKI | Robert-Koch-Institut | Germany's national agency for disease control and prevention |
| ECDC | European Centre for Disease prevention and Control | EU-wide public health body for disease surveillance |
| CW | calendar week | ISO-standard week numbering used throughout the analysis |
| fls | flu season | commonly defined as lasting from CW40 of the starting year to CW20 of the subsequent year (CW40-subsCW20) [6, p. 13,17] |
| $fls_{1,year}$ | flu season 1 of a year | CW04-CW20 of a given year, i.e. second half of an interannual fls |
| $fls_{2,year}$ | flu season 2 of a year | CW40 of a year to CW03 of the subsequent year (CW40-subsCW03), i.e. first half of an interannual fls |
| $sus_{year}$ | summer season | time of the year that is not a fls, i.e. CW21-CW39 |
| $\{a_{coh}, b_{coh}\}$ | model parameters | parameters for exponential fitting of exNAMR curves per cohort |

## 2.2 Data acquisition and curation

It follows a brief synopsis of the German data sources that we based the present analysis and model development on.

**2.2.1 Death data.** Earlier (2000-2020) weekly all-cause mortality count (AMC) data were taken from the Destatis (German office for national statistics) webpage [9]. More current (2021-2024) weekly AMC data were likewise taken from the Destatis webpage [10]. The finest age cohort resolution available on a weekly scale were the fifteen cohorts of '0-29', '30-34', '35-39', ..., '90-94', and '95+' year old. Particularly when comparing time courses of EAM counts (EAMC) with those of mRNA-I and PCR testing (see below), this fine resolution was not meaningful and therefore aggregated to match cohort definitions, yielding the seven age cohorts of '0-29', '30-39', '40-49', ..., '70-79', and '80+' year old.

**2.2.2 Demographic data.** The demographic (age cohort) distribution between 2000 and 2020 (based on the 2011 census) and its presumable distribution onwards (based on the most plausible scenario, the default model variant V1) were taken from the Destatis webpage [11]. The finest age cohort resolution available on a yearly scale are the 101 age cohorts '0-1', '1-2',...,'99-100'. The demography within a single year was assumed not to change at all, i.e. stayed constant in our analysis for every week of the corresponding year. This included the omission of potential birthdays within a year [5], which was assumed to play only a minor role within a five-year-cohort-resolution setting.

**2.2.3 PCR-positive test data.** The curation of the PCR-positive data set, likewise provided by the Robert-Koch-Institut (RKI) as "accompanying data to the weekly report" [12], turned out to be challenging. In this report, the time interval ranged from CW12 in 2020 to CW40 in 2023, the age cohorts were '0-4', '5-14', '15-34', '35-59', '60-79', and '80+' years, such that a direct correspondence to the above age cohorts was not straight forward. As a solution, we conducted a spline interpolation given the ages of 2, 10, 25, 47, and 70 years – i.e. close to the specific median values of the presented age cohorts – to obtain estimates for the median points of the six age cohorts above – i.e. 15, 35, 45, 55, 65, and 75 years. Data for the age cohort '80+' were simply kept as presented. In the result section, we refer to 'normalised test incidence' meaning the weekly number of persons tested positive within an age cohort divided by the year-dependent size of this cohort.

**2.2.4 Data on SARS-CoV-2 variants.** The ECDC made relative occurrences of various SARS-CoV-2 variants available [13] between CW01 in 2020 and CW43 in 2023. From this data set, we extracted the values for Germany. No age cohort resolution was available and thus the distribution of variants was assumed to be independent of age, which, although reasonable, might pose a limitation to our study. In this work, we distinguished between the four variants 'alpha' (B 1.1.7), 'delta' (B.1.617.2), 'omicron' (B.1.1.529, XBB.1.5, XBB.1.5+F456L, BA.1, BA.2, BA.2.75, BA.4, BA.5, BQ.1), and 'others' ('beta' (B.1.351), 'epsilon' (B.1.427/B.1.429), 'eta' (B.1.525), B.1.616, 'gamma' (P.1), 'theta' (P.3), 'kappa' (B.1.617.1), B.1.620, 'lambda' (C.37), 'UNK', B.1.621).

**2.2.5 mRNA-I data.** Compiling a consistent weekly dataset of mRNA-I rates across age groups was challenging. The RKI initially provided a data file as an appendix to their regular communication "Epidemiologisches Bulletin" [14]. This file covered the period from CW52 in 2020 to CW52 in 2021 and reported first and second dose rates for the age groups '0-17', '18-29', '30-39', ... , '70-79', and '80+' years. However, these data only covered first and second doses and thus no "booster". More recently, the RKI has published mRNA-I data through a GitHub repository [15]. We downloaded all available files on the weekly "Impfquote" (cumulative mRNA-I rates) for Germany and extracted the relevant values from CW37 in 2021 to CW46 in 2023. In doing so, we accounted for changes in age group definitions and in how injection status was reported. For example, the variable *Impfquote_18bis59_voll* (September 2021) was later split into *Impfquote_18bis59_boost1* and *Impfquote_18bis59_boost2* (September 2023).

For our analysis, we grouped the data into three broader cohorts: '0-17' (juveniles), '18-59' (adults), and '60+' (seniors). To align these groups with the finer AMR data resolution, we proceeded as follows. For doses one and two, we combined the '0-17' and '18-29' cohorts into a single '0-29' group. The combination was weighted by the size of each subgroup. The remaining cohorts were used as reported by the RKI. For doses three and four, the '60-69', '70-79', and '80+' cohorts were merged into the seniors group, while the '30-39', '40-49', and '50-59' cohorts were merged into the adults group. The '0-29' group was again derived as a weighted average of the 'juveniles' and 'adults'. From these aggregated values, we calculated weekly mRNA-I rates for doses one to four by taking differences of the cumulative values.

Because of data updates in the RKI files, some weekly rates became negative (e.g., CW39 in 2022; see Fig 6). In the results section, we use the term normalised incidence of mRNA-I to refer to the age-specific weekly mRNA-I rate, normalised by cohort size.

## 2.3 Modelling of excess all-cause mortality counts (EAMC)

The methodology for estimating expected AM counts (exAMCs), and thereby excess AM counts (EAMCs), is based on our earlier work [4] and extended here with improved wealth of detail. As before, EAMC was computed for each age cohort as the difference between the observed all-cause mortality count (AMC) and its corresponding modelled exAMC. The all-cause mortality rate (AMR) was defined as the AMC over a given interval (typically a week, season, or year) divided by the size of the cohort, denoted $N_{\text{coh}}$.

To determine expected mortality, we fitted an exponential decay function to the observed AMR data for each cohort over the period 2000-2019. As a novel feature, we conducted this fit not for yearly AMR data, but for each calender week (CW) separately. To account for short-term fluctuations and possible delay in reporting, a 5-week moving average filter was applied on the raw data. The expected normalised AMR (exNAMR) for a certain CW, denoted $\tilde{r}_{AM,\text{coh}}^{\text{ex}}(t)$, was modelled as:

$$\tilde{r}_{AM,\text{coh}}^{\text{ex}}(t) = (1 - b_{\text{coh}}) \cdot \exp\left(-a_{\text{coh}} \cdot (t - t_0)\right) + b_{\text{coh}}, \qquad (1)$$

where $t_0$ = 2000 is the reference year, and $a_{\text{coh}}$, $b_{\text{coh}}$ are cohort-specific parameters determined via non-linear least squares fitting using the Levenberg-Marquardt algorithm (fit routine, MATLAB v2024b). The value of $b_{\text{coh}}$ can be interpreted as an asymptotical limit for large $t$, whereas the parameter $a_{\text{coh}}$ determines the rate at which $\tilde{r}_{AM,\text{coh}}^{\text{ex}}$ decays (if $b_{\text{coh}} < 1$) or rises (if $b_{\text{coh}} > 1$). As in our prior work, NAMR was normalised to the AMR value in the reference year, i.e. it holds $\tilde{r}_{AM,\text{coh}}^{\text{ex}}(2000) = 1$ allowing comparability across time and cohorts. In contrast to our earlier model however, the present version allows the asymptotic value $b_{\text{coh}}$ to exceed 1, accommodating saturation in older cohorts *above* the level of the year 2000.

Once fitted, the exNAMR curves were multiplied by the cohort's actual AMR in the year 2000 to obtain exAMR $\left(r_{AM,\text{coh}}^{\text{ex}}(t)\right)$, and subsequently multiplied by $N_{\text{coh}}(t)$ to yield exAMC. Subtracting these from observed AMC values provides the EAMC for each cohort and time point. To estimate total exAMC and EAMC for the entire German population, cohort-level values were summed up. This approach further enables projection beyond 2019 by extrapolating the fitted exNAMR functions, using the forecast demographic data.

## 2.4 Confidence intervals and significances

For each cohort and each weekly exAMC estimates calculated above for the years 2000-2024, we determined 95% confidence intervals (CI), using the MatLab routine predint, which account for the fluctuations in the observed AMC data points over the time period of fitting. CI radii, i.e. half of the CIs' width, for the whole population were obtained by taking the square-root of the sum of the squared weighted age-cohort-CI radii, according to the variance sum theorem. Observed AMC that exceeded the CI was termed 'significant over-mortality', while those that fell below the CI were termed 'significant under-mortality (UM)'.

When comparing the relative frequency distributions (plotted in histograms) of values of normalised excess AMR (NEAMR), we used a Welch test to check for significant differences in mean values, as the prerequisites for a $t$-test (equal standard deviation of samples) was generally not met.

## 2.5 Cross-correlation analysis

To explore whether cohort-level AM patterns were temporally associated with PCR-incidence or mRNA-I activity, we conducted a cross-correlation analysis between incidence data and NEAMRs. These analyses were performed separately for each age cohort and within two key time intervals of interest. The first interval (CW04 to CW42 of 2021) corresponds to the dominance of the 'alpha' variant and the early 'delta' phase in Germany, prior to widespread administration of third (booster) mRNA-I doses. The second interval (CW30 of 2021 to CW03 of 2022) encompasses the main 'delta' wave and the height of the booster campaign, while largely excluding the first and second mRNA-I doses.

For each cohort and time interval, we calculated Pearson correlation coefficients between NEAMR time series and the following predictors: (i) weekly PCR-incidence, (ii) weekly mRNA-I incidence (doses 1-4), and (iii) the week-wise product of both incidence types as a proxy for joint exposure. These correlation coefficients were computed as a function of temporal lag (in weeks), with negative lags representing incidence events that occurred *before* observed changes in NEAMR. This reflects a causal hypothesis framework: PCR-incidence or mRNA-I events may precede and potentially influence AM trends, whereas reverse causation (NEAMR driving PCR or mRNA-I incidence) is implausible at the population level.

Interpretation of the cross-correlation follows standard convention: a correlation coefficient of 0 indicates no linear statistical association. Positive values indicate in-phase changes: increases in incidence tend to be followed by increases in NEAMR. Negative values indicate anti-phase changes: increases in incidence tend to be followed by decreases in NEAMR. For example, under the assumption of mRNA-I efficacy, one would expect that increased mRNA-I incidence should lead to a subsequent *decrease* in EM, reflected by negative correlation values at negative lags. Conversely, one might expect a positive correlation at negative lags between NEAMR and PCR-incidence, consistent with SARS-CoV-2 waves being followed by subsequent increases in AM.

It is important to stress that cross-correlation analysis does not by itself establish causation, and results must be interpreted with caution. Spurious correlations can arise in lagged time series if the underlying data are not detrended [16] or prewhitened [17]. Accordingly, the present analysis should be understood as exploratory and hypothesis-generating: strong correlations observed in defined time windows highlight patterns that ought to be investigated using more formal causal frameworks.

# 3 Results

In brief, this section explores historical (2000-2024) mortality patterns. We identify a sequence of atypical, cohort-specific EAM signals that begin in late 2020 and persist through 2024. Below we first elucidate how expected mortality was calculated, then characterise weekly and seasonal NEAMR patterns, and finally compare NEAMR with SARS-CoV-2 incidence and mRNA-I uptake.

Expected AM for each 5-year cohort was estimated by fitting exponential functions to pre-pandemic (2000-2019) data; these modelled baselines yield the age-cohort-specific, normalised EAM rate (NEAMR). We analyse NEAMR at weekly resolution and aggregated by season, and use lagged cross-correlations to explore temporal alignment with PCR-incidence and mRNA-I uptake. Key figures summarising these results are Figs 1–8; sex-specific analyses are provided in Supplementary Material S2 Fig.

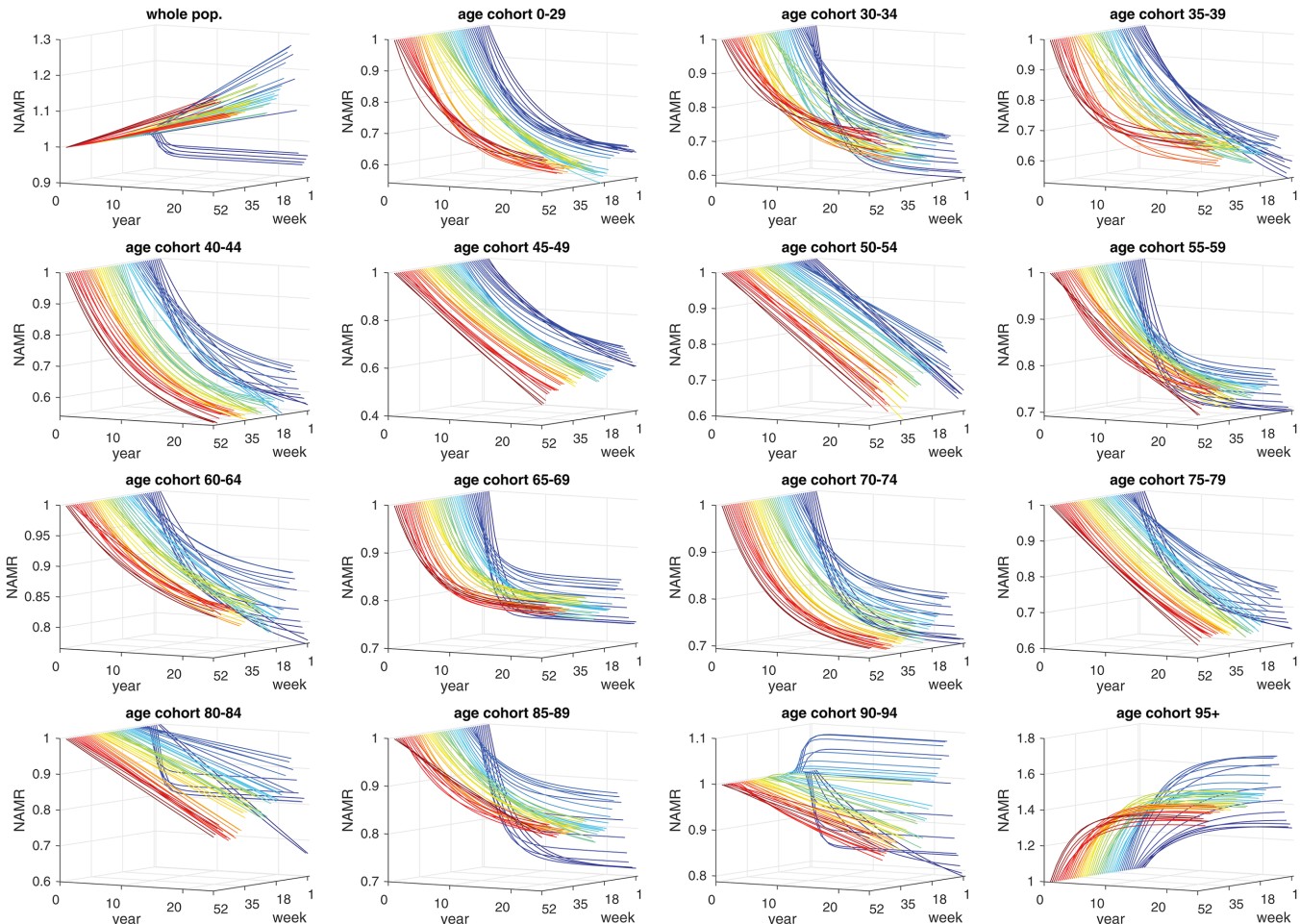

**Fig 1. German expected normalised AMRs (exNAMRs) of the lumped cohort of the youngest (ages 0–29 years), the thirteen older 5-year age cohorts, and the lumped cohort of the eldest (ages 95+).** Plotted are the curves fitted to the observed values of one specific CW in each year, with the observations themselves not shown for clarity of the figure; all observed (year-specific) AMR values were normalised to the CW's value in 2000; for each CW, an exponential two-parameter function (Eq (1)) was fitted through the respective (twenty) observed normalised AMR (NAMR) data points from 2000 through 2019, as depicted by a line with a CW-specific colour ranging from blue (CW01) through violet (CW52); for calculating prognoses (exAMR, exAMC, NEAMR), each CW's fitted exNAMR function was extrapolated beyond 2019. Distinguishing females and males, the same is plotted in Fig A.1 and Fig A.2, respectively.

## 3.1 Weekly exNAMRs

Fig 1 shows the expected normalised all-cause mortality rates (exNAMRs) for 5-year, sex-pooled cohorts over 25 years. These baselines were derived by fitting exponential functions to pre-pandemic NAMR values (2000-2019; see Eq (1)), then extrapolating beyond 2019 [4, Fig 2]. Each 3D cohort panel displays 52 coloured trajectories (one per calendar week) from 2000 onwards. Multiplying an exNAMR value in a given CW of a year by its 2000 exNAMR reference value and the cohort size in that year yields the expected deaths (exAMC) in that CW for that cohort.

Most cohorts follow a clear exponential exNAMR decline over time. The 50-54 cohort shows an almost linear trend, while the eldest groups (90-94 and 95+) exhibit increasing exNAMRs, particularly in late-winter weeks. These baselines define the reference against which observed mortality is compared in subsequent analyses.

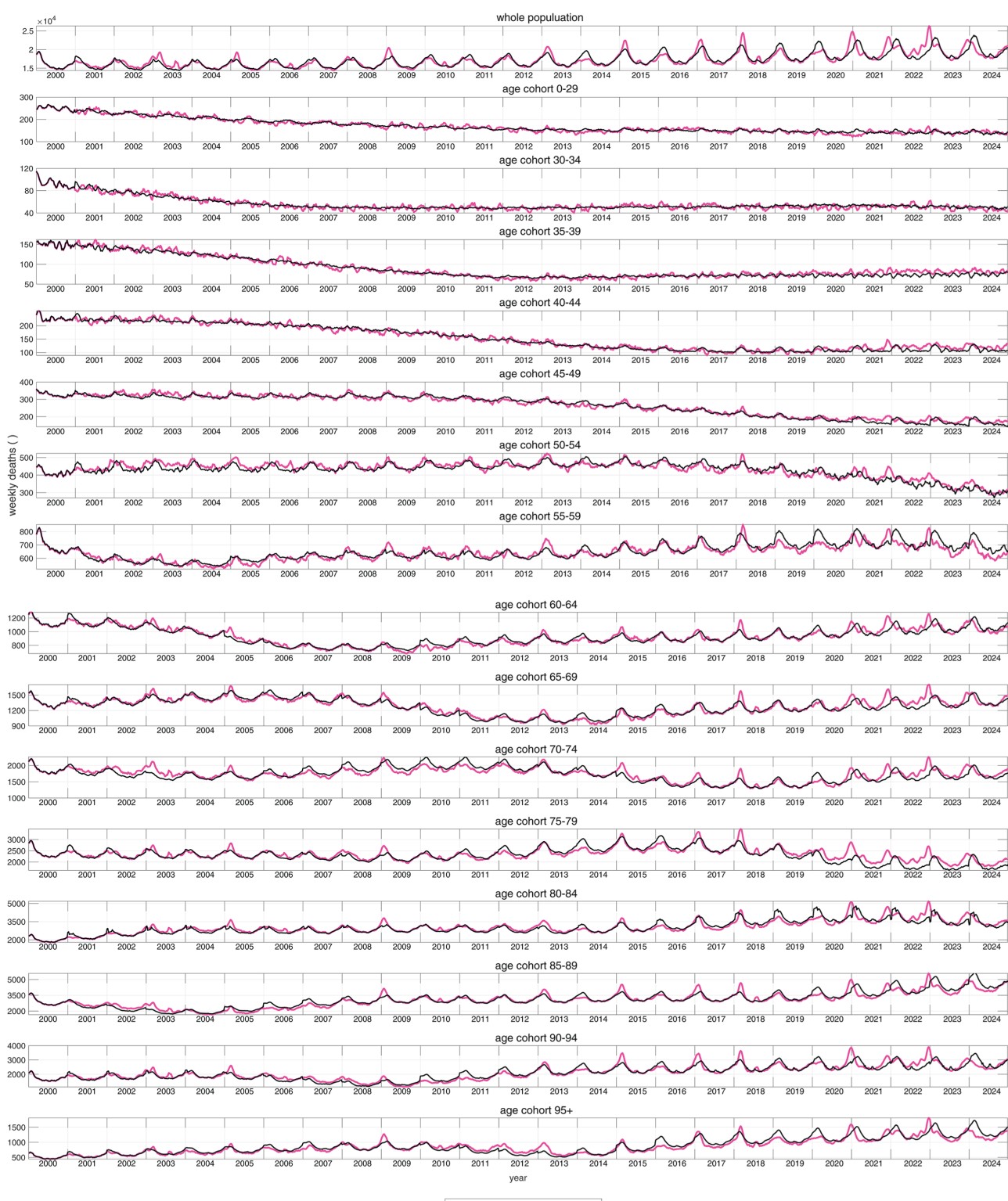

**Fig 2. For the fifteen age cohorts resolving the complete age range, German ($N_{pop} \approx$ 83.6 million in 2024 [18]) weekly observed (magenta lines) all-cause mortality counts (AMCs) and expected (model-estimated), corresponding values (exAMCs; black lines) from 2000 through 2024; an exAMC is the product of its correspondingly fitted exAMR value and the year-specifically observed number of cohort members (demographics; see e.g. [4, Fig 3]).** Distinguishing females and males, the same is plotted in Fig B.3 and Fig B.4, respectively.

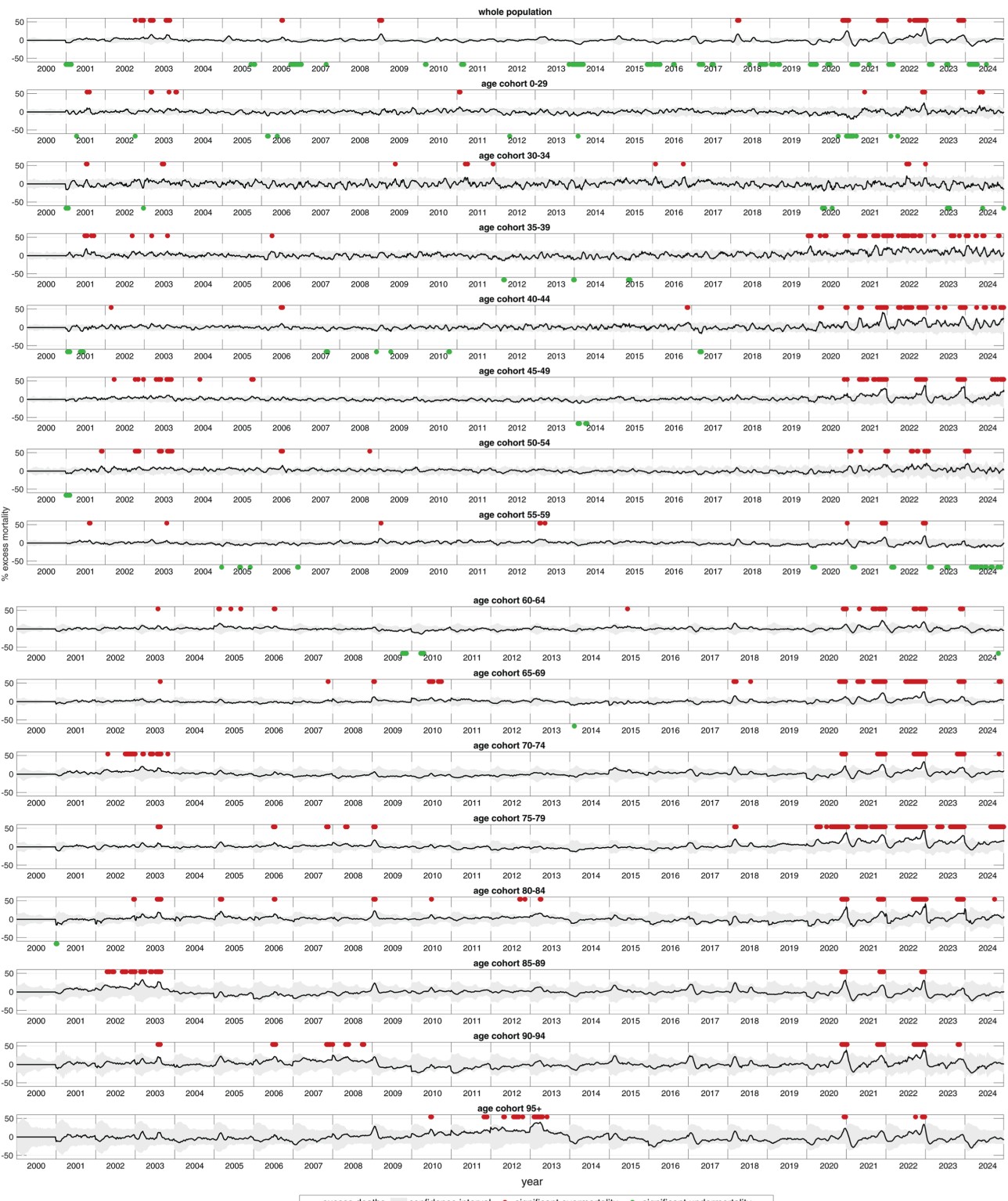

**Fig 3. German weekly normalised excess AMRs (NEAMRs; black lines) from 2000 through 2024, for the same fifteen age cohorts as in Fig 2;** normalisation: weekly excess all-cause mortality count (EAMC: difference between its AMC and corresponding *expected* AMC (exAMC); both: see Fig 2) divided by its exAMC; NEAMR values exceeding (red) or dropping below (green) the 95% CI indicated by a spot. Distinguishing females and males, the same is plotted in Fig C.5 and Fig C.6, respectively.

| year | expAMC | EAMC |
|---|---|---|
| 2000 | 838,448 | -4,582 |
| 2001 | 824,309 | 1,576 |
| 2002 | 815,786 | 23,139 |
| 2003 | 809,753 | 42,526(**) |
| 2004 | 817,527 | -4,384 |
| 2005 | 821,966 | 5,905 |
| 2006 | 828,790 | -9,128 |
| 2007 | 833,498 | -8,795 |
| 2008 | 835,511 | 4,216 |
| 2009 | 846,612 | 4,569 |
| 2010 | 862,466 | -5,891 |
| 2011 | 860,407 | -10,742 |
| 2012 | 870,498 | -5,808 |
| 2013 | 876,724 | 14,808 |
| 2014 | 894,726 | -29,267 |
| 2015 | 911,469 | 12,341 |
| 2016 | 930,430 | -23,835 |
| 2017 | 938,046 | -9,374 |
| 2018 | 948,239 | 4,055 |
| 2019 | 964,734 | -27,950 |
| 2020 | 982,632 | -7,359 |
| 2021 | 992,943 | 25,361 |
| 2022 | 1,002,349 | 60,991(**) |
| 2023 | 1,013,493 | 12,382 |
| 2024 | 1,034,630 | -31,799 |

| season | expAMC | EAMC |
|---|---|---|
| 00/01 | 544,316 | -8,864 |
| 01/02 | 536,958 | 5,589 |
| 02/03 | 532,318 | 33,253(**) |
| 03/04 | 534,113 | -2,574 |
| 04/05 | 538,075 | 7,909 |
| 05/06 | 542,180 | -9,188 |
| 06/07 | 546,094 | -15,519 |
| 07/08 | 548,374 | 5,485 |
| 08/09 | 553,866 | 15,208 |
| 09/10 | 563,659 | -13,571 |
| 10/11 | 567,296 | -11,053 |
| 11/12 | 571,347 | -2,577 |
| 12/13 | 576,878 | 22,890(**) |
| 13/14 | 586,351 | -31,923(**) |
| 14/15 | 598,299 | 18,289 |
| 15/16 | 610,878 | -28,210(**) |
| 16/17 | 619,263 | 7,258 |
| 17/18 | 625,860 | 8,942 |
| 18/19 | 635,716 | -29,497(**) |
| 19/20 | 647,699 | -28,562 |
| 20/21 | 656,826 | 13,303 |
| 21/22 | 663,690 | 21,985 |
| 22/23 | 671,067 | 40,709(**) |
| 23/24 | 683,167 | -12,236 |
| 24/25 | 693,285 | -13,490 |

| season / age cohort | $fls_{1,20}$ | $sus_{20}$ | $fls_{2,20}$ | $fls_{1,21}$ | $sus_{21}$ | $fls_{2,21}$ | $fls_{1,22}$ | $sus_{22}$ | $fls_{2,22}$ | $fls_{1,23}$ | $sus_{23}$ | $fls_{2,23}$ | $fls_{1,24}$ | $sus_{24}$ | $fls_{2,24}$ |
|---|---|---|---|---|---|---|---|---|---|---|---|---|---|---|---|
| whole pop. | -6.3 | -1.7 | 11.5 | -6.2 | 0.6 | 11 | -3.3 | 7.8 | 17.7 | -3.9 | -1.2 | 7.6 | -9.7 | -1.4 | 3.3 |
| 0-29 | -6.3 | -2.8 | -9.5 | -10.8 | 3.9 | 1.2 | -7 | 6.4 | 7.1 | -0.3 | -0.9 | 3 | 1.2 | 5 | -1 |
| 30-34 | -9.3 | -6.5 | -7.6 | -6.3 | -2.6 | -0.6 | -2.7 | 3.8 | 3.5 | -5.3 | -3.7 | -1.1 | -4.8 | -6.9 | -6.3 |
| 35-39 | 7.3 | 3.9 | 8.5 | 8.8 | 10.9 | 12.7 | 11.6 | 17 | 10.7 | 11 | 10.8 | 9.2 | 14 | 7.7 | 5.5 |
| 40-44 | 0.1 | 2.3 | 5.9 | 3.9 | 3.9 | 18.5 | 5.2 | 12.5 | 15.9 | -0.5 | 7.8 | 14.9 | 3.9 | 8.1 | 13.3 |
| 45-49 | 0.1 | 3.5 | 6.3 | 4.5 | 10.6 | 17.8 | -1 | 4.4 | 16.8 | -0.5 | 3.5 | 18 | -3.3 | 5 | 14.9 |
| 50-54 | -2.8 | -2.5 | 1 | 4.3 | 2.4 | 11.4 | 1.1 | 7.1 | 12.3 | 4.2 | 2.5 | 8.3 | 4.8 | 0.4 | 1.6 |
| 55-59 | -8.2 | -3.4 | 0.6 | -5 | 0.3 | 5.7 | -8.3 | -0.7 | 4.4 | -8.6 | -5.8 | -2.6 | -10 | -8.2 | -7.5 |
| 60-64 | -3.2 | -0.9 | 4.7 | -0.5 | 2.7 | 11.6 | -2 | 5 | 9.2 | -1.6 | -2.2 | 3.7 | -5.9 | -3 | -3.4 |
| 65-69 | -0.9 | 0.7 | 6.8 | 2.7 | 3.9 | 12.6 | 0.6 | 8 | 15 | 1.3 | 1.8 | 6 | -3.8 | -0.4 | 2.8 |
| 70-74 | -8.9 | -4.7 | 8.7 | -2.5 | 3.1 | 13.7 | -1.7 | 8.6 | 18.7 | 0.4 | 3.8 | 11.4 | -3 | 3.1 | 7 |
| 75-79 | 8.9 | 10.8 | 21.6 | 10.9 | 13.3 | 20.4 | 5.4 | 19.1 | 26.9 | 6 | 11.6 | 18.6 | -1.8 | 9.3 | 13.7 |
| 80-84 | -10.7 | -6.3 | 11.6 | -9.6 | -3.5 | 11.6 | -3.4 | 7.6 | 21.5 | -1.7 | 0.4 | 14 | -4.9 | 3.3 | 8.4 |
| 85-89 | -9.8 | -4.4 | 11.5 | -13.4 | -5.7 | 5.8 | -10.2 | 1.7 | 12.1 | -11.7 | -8.9 | 1.5 | -16.5 | -7.2 | -1 |
| 90-94 | -7 | 0.4 | 18.4 | -9.6 | 2.2 | 12.9 | -0.9 | 12.7 | 25 | -3.8 | -0.3 | 8.7 | -16.3 | -3.8 | 3.7 |
| 95+ | -13.1 | -4.1 | 7.8 | -16.5 | -2.4 | 4.7 | -5.6 | 6.8 | 15.8 | -12 | -8.7 | -1.5 | -20.1 | -8.8 | -4 |

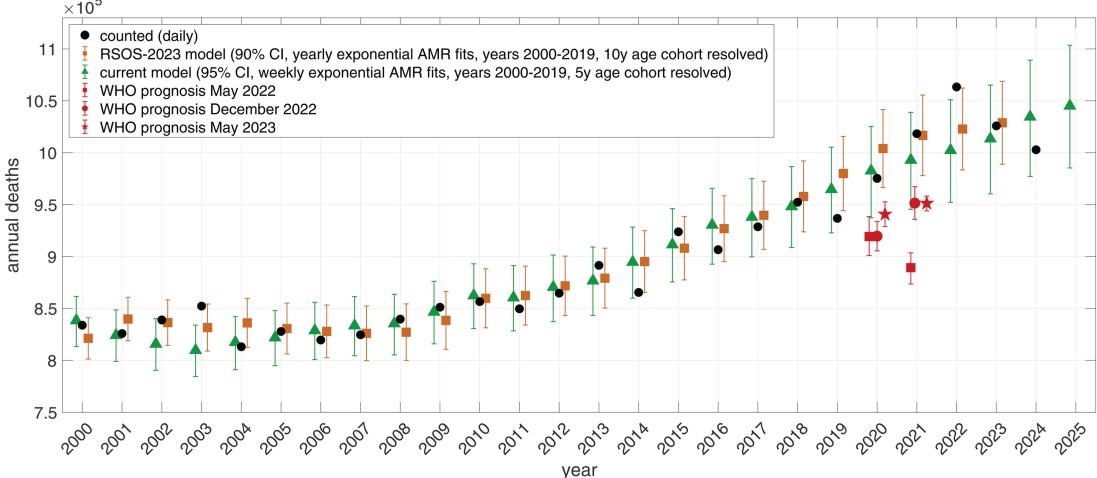

**Fig 4. Top: Yearly AMCs** (i.e. observed: black dots) **and exAMCs** (i.e. expected: green triangles enclosed by 95% CIs), respectively, for the years 2000-2024 in Germany for comparison, exAMCs estimated by our original model version [4] (yearly instead of weekly NAMR resolution, 10-year instead of 5-year age cohorts: orange squares enclosed by 90% CIs); model population in 2024: $N_{pop}$ = 83.6 million [18]; Various WHO predictions (red symbols) are shown for comparison, see [4, Fig 4] for a more detailed explanation. **middle, left:** yearly values of exAMCs and EAMCs (UM if negative); **middle, right:** the same for just the 'flu seasons' ($fls_{2,year}$ & $fls_{1,year+1}$ : CW40-subsCW20); 95%-CI significance of EAMC indicated by (**); **bottom:** seasonal percentage NEAMR values of the 5-year-resolved age cohorts since 2020 through 2024.

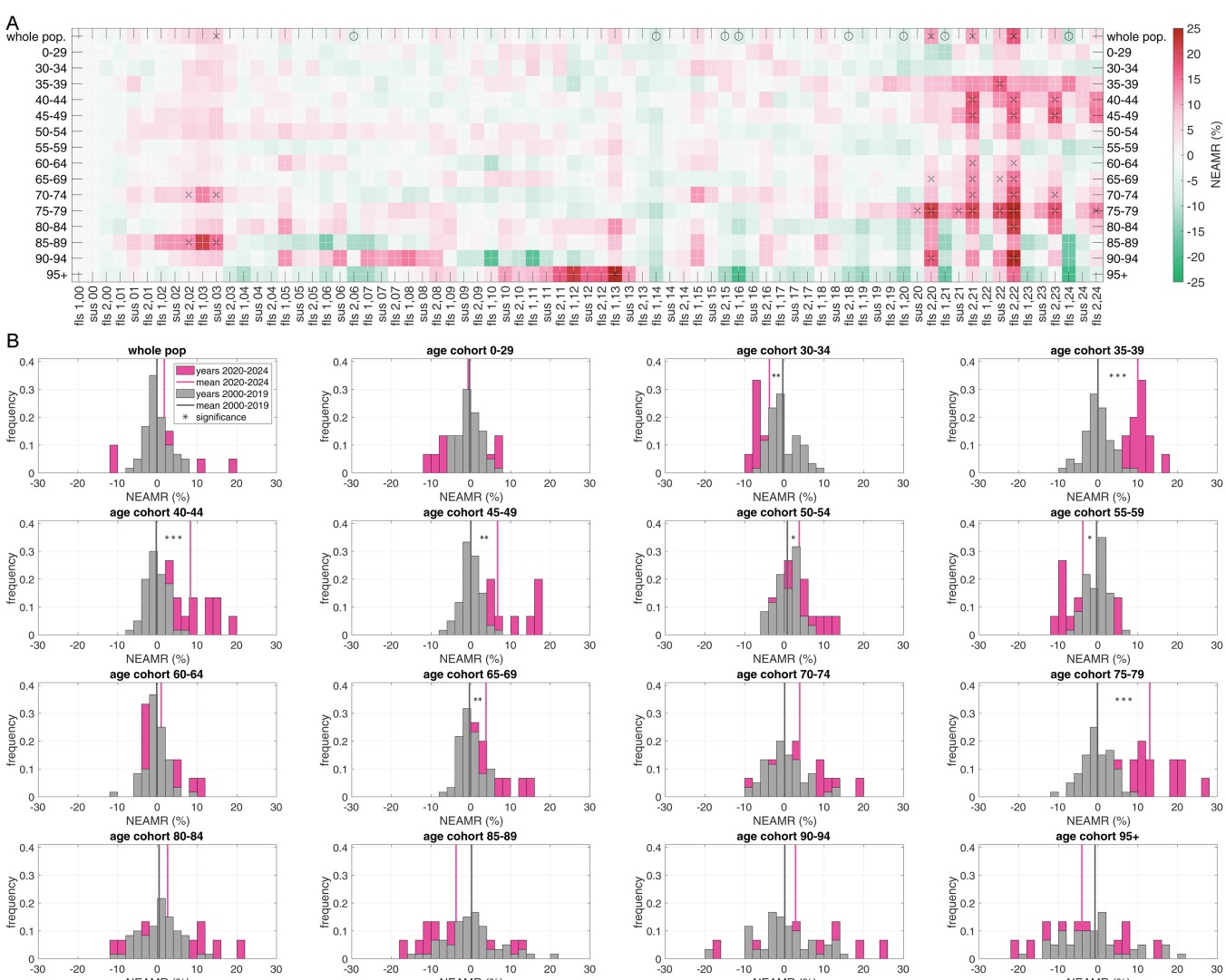

**Fig 5. German seasonal NEAMRs from 2000 through 2024, for the same fifteen age cohorts as in Figs 2, 3; NEAMR values given in % as in Figs 3, 6, however, time-resolved more coarsely for just three seasons constituting a full year (A):** a tile is plotted for each of the three seasons of any year ('flu season 1', fls$_1$: CW04-CW20; 'summer season', sus: CW21-CW39; 'flu season 2', fls$_2$: CW40-subsCW03), which codes by colour the percentage NEAMR values averaged in time over the respective season (reddish: over-mortality, i.e. positive NEAMR; greenish: UM, i.e. negative NEAMR), now indicating significance by either an asterisk (NEAMR exceeding 95% CI) or a circle (NEAMR dropping below 95% CI); B: two distributions of the relative frequencies (histograms) of NEAMR values are plotted in one sub-panel, of which each reflects the data of one age cohort: one distribution for the years 2000-2019 (grey), and one for 2020-2024 (magenta); their arithmetic mean values are symbolised by solid vertical lines; the significance of the difference in mean values is indicated by star symbols: $p < 0.05$ (one), $p < 0.01$ (two), $p < 0.001$ (three); distinguishing females and males, the same is plotted in Fig D.7 and Fig D.8, respectively.

## 3.2 Weekly AMCs, exAMCs, and NEAMRs from 2000 through the 'first Corona wave' in early 2020

Fig 2 compares observed weekly mortality counts (AMCs, magenta) with expected values (exAMCs, black) across cohorts from 2000 to 2024. The total-population exAMC (top panel) shows a pronounced, sawtooth-shaped seasonal pattern with sharp winter peaks, minor summer fluctuations, and gradual structural shifts since 2010.

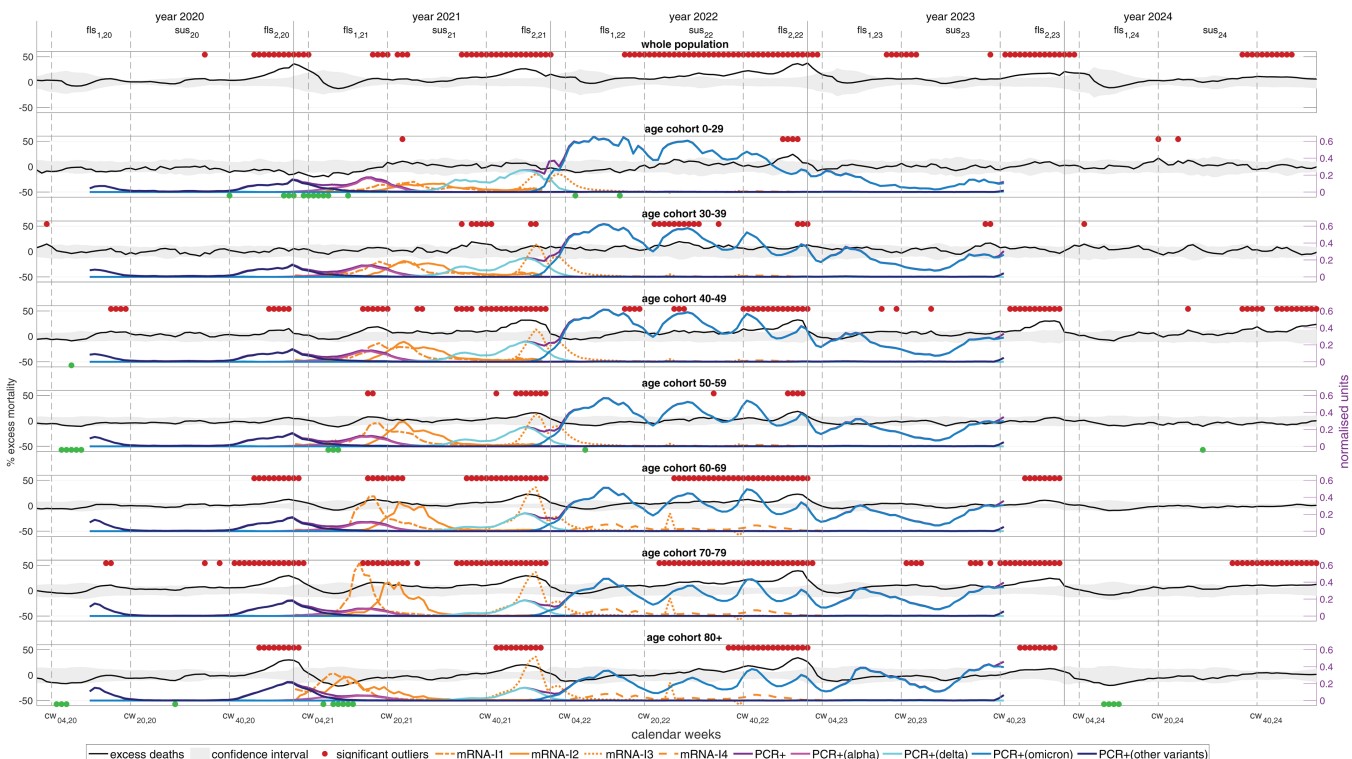

**Fig 6. German weekly NEAMRs (black lines) from 2020 through 2024, however, with coarser age resolution (see Sect 2.2.5) of only seven instead of the fifteen cohorts applied in Figs 2, 3; 95% CI is the grey shaded area; NEAMR values exceeding (red) or dropping below (green) the 95% CI are indicated by a spot; abscissa: CW#; ordinates: left for NEAMR values (given in %; see Fig 3), right for all others (normalised incidence values). Additionally shown: (i; orange lines; 'normalised mRNA-I incidences', mRNA-I#) weekly** *number of mRNA-I*, **normalised to the cohort's size, and multiplied by 25 for better visibility, i.e. adopted to the ordinate range of (ii), distinguishing the first (mRNA-I1), second (mRNA-I2), third (mRNA-I3), and fourth (mRNA-I4) mRNA-I. (ii; violet, magenta, cyan, sky blue and dark blue; 'normalised PCR-incidences', PCR+) weekly** *number of positive PCR tests*, **normalised to the number of PCR tests conducted as well as to (times) the cohort's size, the numbers distinguishing 'alpha' (magenta), 'delta' (cyan), 'omicron' (sky blue), and 'others' (dark blue), as well as the total number not distinguishing SARS-CoV-2 variants (violet).** Distinguishing females and males, the same is plotted in Fig E.9 and Fig E.10, respectively.

Condensing these data into NEAMR (Fig 3) highlights major mortality anomalies. Red points indicate weeks of NEAMR exceeding the 95% CI (grey area); green points indicate weeks of NEAMR falling below it. A double peak in 2003 (February and July) affected nearly all older cohorts, while winter flu seasons in 2008/09, 2012/13, 2014/15, 2016/17, and 2017/18 showed marked EAM in those over 50. In contrast, the 2018/19 and 2019/20 seasons lacked the usual winter peaks in older groups.

At the start of 2020, EAM was minimal. Apart from a brief signal in the 35-39 cohort around New Year and in 75-79 year olds during March, no significant EAM was observed in other cohorts or the population as a whole. The March signal in 75-79 year olds appears to reflect an unusually extended seasonal tail, with observed mortality only slightly above baseline. Thus, the 'first Corona wave' was not associated with substantial EAM in Germany.

### 3.3 Patterns of NEAMR from mid 2020 through 2024

From late 2020 onwards, NEAMR patterns changed fundamentally (Fig 3). At the population level, four consecutive December peaks (2020-2023) were each followed by pronounced UM dips in January or February. At the cohort level, signals varied in timing, magnitude, and

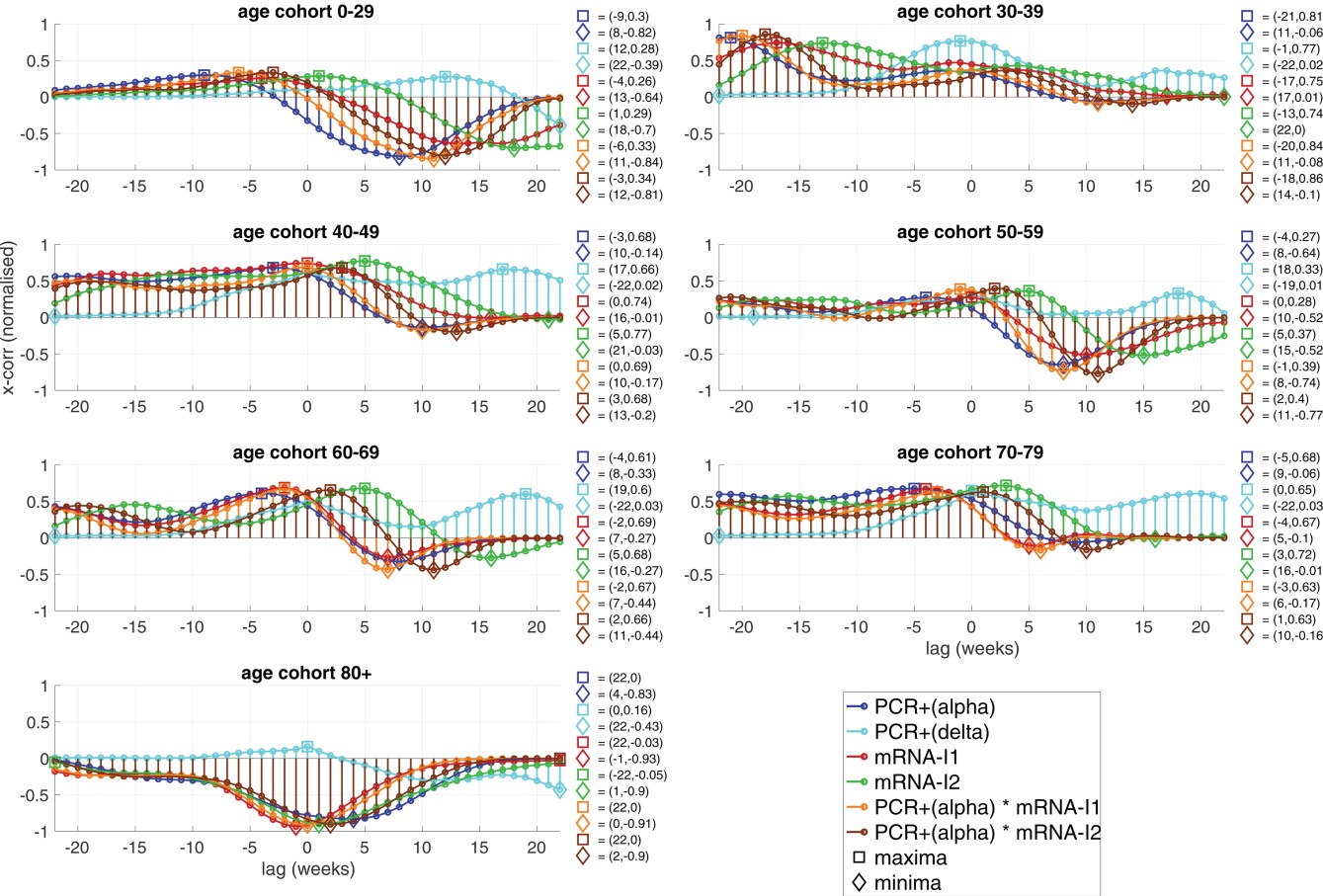

**Fig 7. In Germany, during the time interval CW04-CW42,2021, and for seven age cohorts, the (Pearson) coefficients of cross-correlating their respective weekly NEAMR time course with several (time-lagged) normalised incidence signals (regarding normalisation, see caption of Fig 6) are plotted:** with the *normalised weekly numbers of positive PCR tests* PCR+(V) of two SARS-CoV-2 variants V∈ {'alpha', 'delta'}, and with the *normalised weekly number of mRNA-I* of the first (mRNA-I1), and second (mRNA-I2) mRNA-I, as well as with two products by week of incidences; maximum and minimum coefficient values to the right of a sub-panel; distinguishing females and males, the same is plotted in Fig F.11 and Fig F.12, respectively.

persistence. To aid readability, we summarise findings by cohort group, with each paragraph beginning with a brief synopsis of its main result.

**Whole population: Four December peaks followed by early-year UM dips, with atypical persistence through 2024.** The population-wide NEAMR signal from 2020 to 2023 was highly stereotyped: each year produced a sharp December peak, followed by a pronounced UM dip in January or February (Fig 3, top panel). Peak magnitudes reached about 25% in 2020, a similar level in 2021, the highest at 30% in 2022, and 14% in 2023. The 'first Corona wave' in spring 2020 was insignificant by comparison, with NEAMR within the 95% CI until November. In 2022, EAM rose unusually early, remaining above the CI from July through December. In 2024, the signal stayed mostly within the CI after an early-year dip, leading to overall UM.

Annualised EAMCs corroborate this picture of atypical and persistent EAM (Fig 4, middle panel). They show slight UM in 2020 (–7,359), moderate EAM in 2021 (25,361), and a pronounced EAM peak in 2022 (60,991) comparable to the 2003 SARS-CoV-1 winter surge

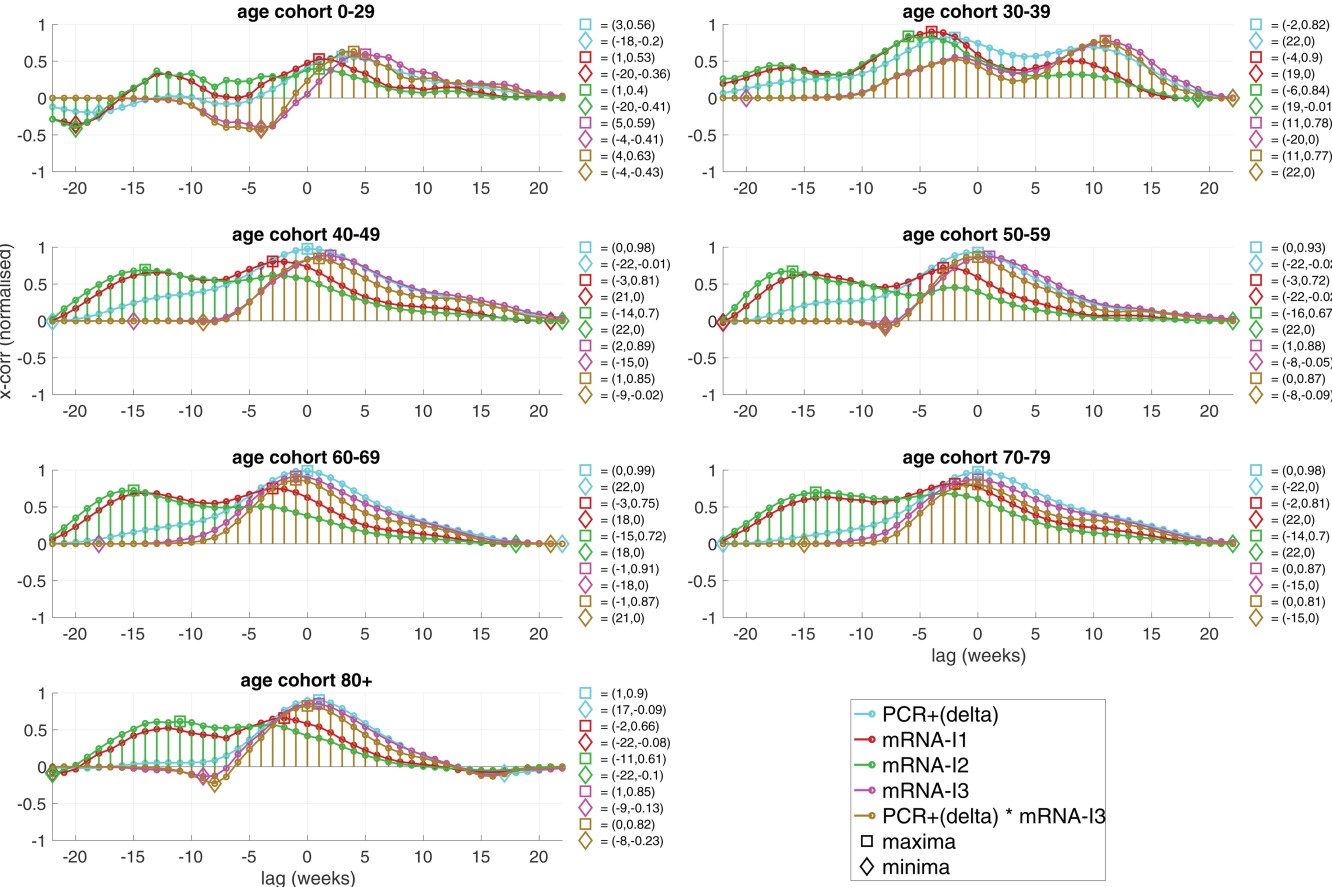

**Fig 8. In Germany, during the time interval CW30,2021-CW03,2022, and for seven age cohorts, the (Pearson) coefficients of cross-correlating their respective weekly NEAMR time course with several (time-lagged) normalised incidence signals (regarding normalisation, see caption of Fig 6) are plotted: with the *normalised weekly numbers of positive PCR tests* PCR+(V) of the SARS-CoV-2 variant V='delta', and with the *normalised weekly number of mRNA-I* of the first (mRNA-I1), second (mRNA-I2), and third (mRNA-I3) mRNA-I, as well as with one product by week of incidences; maximum and minimum coefficient values to the right of a sub-panel; distinguishing females and males, the same is plotted in Fig G.13 and Fig G.14, respectively.**

(42,526). These updated values, based on 5-year age cohorts and weekly resolution, exceed earlier estimates by 12,000-18,000 deaths [4]. The following years then returned to moderate EAM in 2023 (12,382) and considerable UM in 2024 (−31,799).

The December 2020 peak was broadly synchronised across nearly all age cohorts, with exceptions in the youngest (0-34 years) and in females younger than 60 (Fig C.5). A slight delay was also visible in 50-54-year-olds, peaking in CW02 of 2021 (Figs C.5, C.6). The persistence of atypical, often significant EAM after 2020 is most striking in older cohorts (above 60, particularly 75-79) and in younger adults (35-49), but less evident in the eldest (95+), those aged 50-59, and the youngest under 35.

Taken together, these results show that from late 2020 onwards the German population experienced a new mortality pattern characterised by repeated late-year EAM peaks and subsequent early-year UM dips, with atypical persistence into the summer of 2022 and renewed EAM in late 2023.

We now turn towards the NEAMR signals of the age cohorts (panels below top one in Fig 3), starting with the eldest (bottom panel in Fig 3), and running through to the youngest.

**95+: Predominantly, UM with only modest EAM peaks after 2020.** The eldest cohort showed persistent UM from 2014 until summer 2020, then modest December 2020 excess, where the population-wide significant EAM is fully reflected in those 95 and older. However, no major peaks occurred in 2021, only moderate late-2022 signals, followed again by UM throughout 2023-2024. Notably, during the period between late 2011 until early 2013, excessive 95+ EAM occurred in the generation born in world war one; for this generation, corresponding significant EAM signals can be found in 2007/2008 in the 90-94 cohort, and in 2002/2003 (SARS-CoV-1) in the cohort 85-89.

**90-94 and 85-89: Three consecutive December peaks, then UM.** These cohorts closely mirrored the 95+, but with stronger signals. Each December from 2020-2022 showed significant peaks, followed by average or slightly low mortality in 2023 and UM in 2024.

**75-79: Persistent and severe EAM since early 2020.** This cohort was the most heavily affected during the SARS-CoV-2 era. Significant EAM persisted almost continuously from March 2020 to late 2024, interrupted only by short UM dips in early 2021-2023 and a longer pause in early 2024. Later in 2024, mortality rose again, though as a sustained signal rather than sharp peaks.

**70-74 and 80-84: Strong winter peaks including late 2022 and 2023.** These groups resembled each other, with major November-to-December peaks in 2020-2022 and additional winter signals in 2022. Their EAM periods were shorter than in 75-79 but longer than in the oldest cohorts.

**65-69: Consistently, high winter peaks, second only to 75-79.** This cohort experienced strong December EAM in 2020-2023, with signal lengths between those of 75-79 and 70-84. It was the second-most affected group among the elderly.

**60-64: Moderate excess with unique spring peaks.** Patterns resembled 70-74 and 80-84, though with weaker late-2024 EAM signals. Unique features included a spring 2021 excess and prolonged late-2021/early-2022 periods above baseline.

**55-59: Largely spared, with frequent UM dips.** This group was the least affected. Only short December EAM signals were seen in 2020-2022. Each year from 2020-2024 also displayed significant early-year UM, and 2024 showed almost persistent UR.

**50-54 and younger adults: Generally, low EAM with sporadic fluctuations.** The 50-54 cohort showed modest EAM peaks scattered across 2021-2024. The 30-34 cohort remained largely unaffected, with only isolated signals in 2022. These younger groups were dominated by higher-frequency fluctuations rather than the clear annual sawtooth patterns of older cohorts.

**45-49: Increasingly severe peaks, especially in 2023.** EAM was minor in 2020 but rose in later years, culminating in the strongest peak of any cohort in December 2023. A gradual increase throughout 2024 produced further late-year EAM.

**35-39 and 40-44: Persistent and pronounced EAM since 2021.** Both cohorts displayed strong EAM from 2021 onward, with peaks in March 2020, late 2020, and December 2022, followed by renewed EAM in late 2023 and throughout 2024. These cohorts, together with 75-79, were among the most heavily affected.

**0-29: Unusual UM-EM alternations after 2020.** This youngest group showed atypical dips into UM in late 2020/21 and again in late 2021/22, followed by persistent EAM in mid-to-late 2022 and a prolonged EAM phase through much of 2024. Despite relatively small magnitudes, the alternation between deep UM and extended EM is a distinctive feature.

Next, we examine these persistent mortality phases more systematically by averaging NEAMR over three seasonal intervals per year (about 17 weeks each), enabling comparison of seasonal-level changes across cohorts.

## 3.4 Seasonal NEAMRs

Seasonal averaging highlights broad changes in mortality patterns while narrowing down week-to-week variability (Fig 5). At the whole-population level, only one summer season before 2020 showed significant EAM ($sus_{03}$), whereas several flu seasons between 2000 and 2019 showed UM. By contrast, 2020-2024 was characterised by extremes in both directions: three consecutive winter seasons (2020-2022) with pronounced EAM, balanced by strong UM in corresponding flu seasons. Overall, average EAM across the two periods (2000-2019 vs. 2020-2024) was not significantly different.

The histogram for the whole population (bottom panel of Fig 5, top-left sub-panel) highlights two main observations: (i) average EAM in 2020-2024 is not significantly higher than in 2000-2019; and (ii) two seasonal peaks (both $fls_2$) in 2020-2024 exceeded any EAM in the prior 20 years, while two other seasons (both $fls_1$) showed more extreme UM than any earlier season. These extremes effectively balance each other out, rendering the overall difference between the two periods statistically insignificant.

Cohort-specific analyses reveal striking contrasts hidden at the population level. The large summer 2003 peak originated in older groups, particularly 70-74 and 85-89 years, whereas the recent winter peaks (2020-2022) involved nearly all cohorts. Exceptions included the youngest (0-29 and 30-34), the robust 55-59 group, and the oldest cohorts (85-89 and 95+), whose seasonal mortality distributions showed no significant shift relative to pre-2020 baselines.

By contrast, several middle-aged and elderly cohorts exhibited clear, statistically significant changes. Mortality increased strongly in 65-69 ($p < 0.01$) and 75-79 ($p < 0.001$) year olds, with the latter providing the largest contribution to the whole-population EAM. Equally noteworthy, adults aged 35-49 (typical age of school-child parents) showed unprecedented seasonal EAM since late 2020, with significance levels ranging from $p < 0.001$ (35-44) to $p < 0.01$ (45-49), and $p < 0.05$ (50-54). These shifts underline that both older and middle-aged cohorts were disproportionately affected after 2020.

Finally, a few groups deviated in the opposite direction. The 30-34 and 55-59 cohorts showed weak but statistically significant UM ($p < 0.05$) in 2020-2024 compared with the earlier period. This emphasises that EAM in Germany after 2020 was not a uniform population-wide phenomenon but the result of heterogeneous age-specific patterns.

## 3.5 Weekly courses of NEAMR, PCR-positive tests, and mRNA-I in 2020-2023

Fig 6 overlays weekly NEAMR signals with PCR-incidence and mRNA-I uptake. This comparison highlights how mortality dynamics aligned with both incidence waves and injection campaigns across different age groups.

PCR-incidence captures the sequence of infection waves: the first (March 2020), the second (late 2020), the third ('alpha', spring 2021), and two subsequent 'delta' waves (summer and autumn 2021). Incidence peaked at comparable levels across younger and middle-aged groups but was increasingly lower in older cohorts, especially during 'alpha'.

The mRNA-I campaign followed closely on these waves. The first two mRNA-I doses were largely complete by July 2021. As the booster campaign (third dose) began in autumn 2021, 'delta' incidence peaked simultaneously, particularly in younger cohorts. Shortly thereafter, 'omicron' emerged in coincidence with the booster peak in December 2021.

The 'omicron' period (2022-2023) was marked by extremely high PCR-incidences, reaching up to 60% in some age groups before reporting ceased in October 2023. Four major PCR-incidence waves occurred during 2022, followed by two smaller ones in 2023, closely aligned with both PCR-incidence and mRNA-I activity.

To determine whether the observed temporal similarities between NEAMR, PCR-incidence, and mRNA-I activity represent random coincidence or structured lagged association, we next employ cross-correlation analysis. This analysis will serve as both an exploratory tool to test temporal dependency and as support for an immunological hypothesis (see Sect 4.1).

### 3.6 Cross-correlations between NEAMR and SARS-CoV-2 incidence and mRNA-I data

Figs 7 and 8 summarise cross-correlations between NEAMR and weekly incidence of PCR-incidence, mRNA-I uptake, and their product terms. We focus on negative lags, reflecting incidence events that preceded mortality outcomes.

**CW04-CW42 of 2021: 'alpha' to early 'delta'.**   During this first interval, correlations were heterogeneous across cohorts. Among the eldest (80+), initiation of the first mRNA-I dose (mRNA-I1) was strongly associated with subsequent reductions in mortality (peak coefficient -0.93 at lag -1 week). In younger cohorts, correlations were weaker or positive. In 30-39 year olds, however, a clear signal emerged: NEAMR correlated strongly with combined second-dose mRNA-I and PCR-('alpha'-)incidence (mRNA-I2×PCR+(alpha), coefficient 0.86 at lag -18 weeks). Correlations with mRNA-I1×PCR+(alpha) (0.84, lag -20) and PCR+(alpha) alone (0.81, lag -21) showed similar timing, while 'delta' incidence correlated more acutely (0.77 at lag -1). These patterns suggest a tight temporal alignment of mortality changes in this group with the first mRNA-I campaign and the transition from 'alpha' to 'delta'.

**CW30 of 2021-CW03 of 2022: 'delta' dominance and booster campaign.**   In this second interval, correlation patterns became more uniform. Across all cohorts aged 40+, NEAMR correlated very strongly with 'delta' incidence, with coefficients of 0.93-0.99 peaking at or near zero lag. For 30-39 year olds, the peak correlation was moderate (0.82 at lag -2), while the youngest cohorts showed no substantial correlation. The booster campaign (mRNA-I3), initiated in CW40-45, coincided with peak 'delta' incidence. NEAMR correlated almost identically with mRNA-I3 incidence and with the combined mRNA-I3×PCR+(delta) variable, both peaking at lags of 0 to -1 weeks. This reflects a close overlap in timing between rising mortality, high PCR-incidences, and booster uptake. Positive correlations with mRNA-I1 and mRNA-I2 were also present, though weaker (typically <0.73), except again in 30-39 year olds, where coefficients reached 0.9 (mRNA-I1 at lag -17) and 0.84 (mRNA-I2 at lag -6).

Taken together, the cross-correlations show that mortality changes in cohorts older than 30 years were closely aligned with both PCR-incidence waves and mRNA-I campaigns during 2021. The strongest signals occurred in the 40+ cohorts during the 'delta' period, but younger adults also exhibited distinct correlations linked to the timing of injection and variant transition.

## 4 Discussion

### 4.1 Prolonged population-wide PCR-incidence and rising sick leave: an immunological hypothesis

The year 2022 marked an epidemiologically abnormal phase in the course of the SARS-CoV-2 era in Germany. With the rise of the 'omicron' variant, the normalised PCR-incidence rose to persistently high levels across all age cohorts (Fig 6). Weekly PCR-incidence ranged between approximately 0.4 (40%) in the youngest cohort and 0.2 (20%) in the eldest – levels that exceeded any previous peaks observed during the second wave (December 2020), the 'alpha' wave (spring 2021), or the 'delta' wave (autumn 2021). Remarkably, this unprecedented and

enduring PCR-incidence began precisely as the nationwide mRNA-I3 ('booster') campaign reached its peak in early December 2021.

We propose a hypothesis that connects this sustained PCR-incidence to a broader population-level immune dysregulation. Specifically, we suggest that the cumulative effects of repeated mRNA-I – with approximately three doses administered to the majority of the adult population by the end of 2021 – may have temporarily impaired innate immune resilience, increasing susceptibility to both SARS-CoV-2 and other pathogens. This hypothesis is supported by a concurrent surge in non-CoViD-19 illness: Official figures from Destatis report a sharp rise in average annual sick days per employee from 11.2 in 2020 to 14.8 in 2022 and 15.1 in 2023, exceeding the previous historical maximum of 13.0 days in 1995 [19]. The AOK public health insurance consortium, covering nearly 30 million Germans, confirms this trend, reporting a surge in sick days from 19.7 in 2021 to 24.5 in 2022, and stably elevated values (23.9) in 2023 and 2024 [20,21]. While numerous factors may have contributed to this rise, the coincidence of widespread mRNA-I, elevated sick leave, and persistent high PCR-incidence suggests that immunological mechanisms merit closer examination.

Recent findings in molecular immunology provide plausible biological underpinnings for such an effect. Notably, studies have demonstrated that repeated exposure to SARS-CoV-2 spike protein (SP) – whether through infection or mRNA-I-induced expression – can epigenetically reprogram innate immune cells. Simonis et al. [22], for instance, showed that macrophages derived from mRNA-I-treated individuals exhibited persistent priming marked by histone H3K27 acetylation (H3K27ac) – an epigenetic marker of gene activation. Their study revealed a cumulative effect: three mRNA-I induced nearly nine times the number of acetylated promoter regions compared with one mRNA-I, with these changes persisting for at least six months. These modifications, associated with the activation of the NLRP3 inflammasome, predispose the immune system to heightened inflammatory responses upon subsequent stimulation.

This concept of trained immunity – while beneficial in some infectious contexts – may, under repeated or persistent antigen exposure, lead to dysregulation. Chronic activation of the NLRP3 inflammasome, e.g. by repeated SP presentation, has been linked to inappropriate cytokine release (e.g., IL-1$\beta$ [23]), increased systemic inflammation, and immune fatigue. Repeated presentation [23, Fig 4M] of the SARS-CoV-2 SP by mRNA-I in particular stimulates even prolonged and thus accumulative cytokine release, because any mRNA-I induces sustained SP production by body cells. Importantly, the same epigenetic signatures may reduce the effectiveness of the mucosal barrier by down-regulating protective antibodies such as IgA [24,25], and can shift the balance toward inhibitory IgG4 antibodies that may impair long-term antiviral defence [26,27].

Evidence from observational studies lends support to this interpretation. Some large-scale analyses suggest that individuals who received recent mRNA-I experienced an increased risk of SARS-CoV-2 infection compared with those mRNA-I-treated earlier [28–30]. While confounding cannot be excluded, these findings are consistent with a temporary window of heightened vulnerability, potentially reflecting immune modulation rather than enhancement.

Lastly, the detection of SP or its fragments in circulation for extended periods – reportedly up to several months, and in some cases over a year [31] – raises additional concerns. Sustained expression of viral antigens within host tissues can be expected to act as a persistent, and thus fatiguing, immune stimulus, especially if SP production occurs in an uncontrolled or tissue- or organ-specific manner.

Taken together, the available epidemiological and molecular evidence suggests that the widespread deployment of mRNA-I had and continues to have unintended effects on immune

homeostasis in a subset of the population. These effects could plausibly have contributed to prolonged PCR-incidence during 2022 and beyond, and contribute to the observed sustained rise in general illness. Future studies should rigorously investigate these relationships using prospective, immunologically detailed cohort data.

## 4.2 Patterns of persistent EM and UM across generations

The emergence of persistent EAM patterns in specific age cohorts, both during and outside of the SARS-CoV-2 pandemic period (in Germany, officially declared to end on April 7th, 2023), raises important epidemiological and sociomedical questions. Most notably, we observed significant EAM from autumn 2021 onward in the age groups typically associated with parenthood of school-aged children, particularly the 40-49 year old and, even more distinctly, the 35-39 year old. These patterns are not only temporally consistent, but also statistically robust across multiple flu seasons (fls$_2$ of 2021 through 2024), with NEAMR values persistently exceeding the 95% confidence interval (Fig 5).

What makes these findings more striking is their historical uniqueness: prior to autumn 2021, these cohorts exhibited only moderate fluctuations in mortality and were never among those contributing significantly to population-wide EM. We interpret this sudden and prolonged mortality signal as multifactorial. On the one hand, as elaborated in Sect 4.1, immunological stress induced by repeated mRNA-I – particularly when administered within a short interval during ongoing viral circulation – may have compromised the immune robustness of these adults. On the other hand, this cohort also faced extraordinary psychosocial burdens due to school closures, remote working, and familial stress during the pandemic period [32,33]. The resulting chronic stress may have acted as a co-factor, weakening physiological resilience and thereby contributing to elevated AM [34].

Moreover, the most significant and continuous EAM is not seen in the 40-44 cohort but rather in the adjacent 35-39 group (histogram in Fig 5 in particular), which exhibits sustained EAM even during non-winter seasons, such as summer 2022 – an unusual period for elevated AM outside of heatwave events. While EAM among 40-49 year old appears more oscillatory and is partially concealed at the population level due to their relatively lower absolute AMCs, the persistence and magnitude of NEAMR signals in both groups render this a demographically and epidemiologically exceptional phenomenon.

This current mortality anomaly invites comparison with earlier patterns in elderly cohorts. For instance, strong EAM signals were observed in 2003 among the 85-89 and 70-74 age groups. Subtracting their average age from the year, i.e. $2003 - 87 \approx 1916$ and $2003 - 73 \approx 1930$, points to cohorts born during World War I and those who were both born at the beginning of the Great Depression plus had to spend their childhood during World War II, respectively. These are generations likely shaped by early-life stressors, malnutrition, or trauma – factors known to influence long-term health trajectories [35–38]. Five years later, in 2008, the same individuals had shifted into the 90-94 and 75-79 year old cohorts, respectively, where again elevated AM was observed (albeit slightly below statistical significance), and again in 2012-2013, when this same war-affected generation reached 95+ years of age, EAM was prominent and, in parts, statistically significant. These longitudinal cohort traces suggest that early-life adversity may leave a lasting biological imprint that manifests as increased vulnerability in later life. While speculative, such a pattern aligns with existing theories of developmental origins of health and disease (DOHaD), which posit that prenatal and early childhood environments have long-term impacts on immunity, metabolic health, and life expectancy [37,38].

During the SARS-CoV-2 era, another cohort – the 75-79 year old – showed remarkable and persistent EAM throughout 2020-2024, with almost no return to UM in any of the five flu seasons. This cohort was born between 1941 and 1949, during and immediately after World War II. They would have spent part of their early childhood and formative years in post-war Europe – a time of reconstruction, nutritional scarcity, and psychological upheaval. The persistence and magnitude of their EAM signals suggest that they, too, represent a demographically vulnerable population, either due to physiological ageing, life-course exposures, or possible interactions with pandemic-era interventions, including mRNA-I.

In contrast, two age groups stood out for their consistent UM: the 30-34 year old and the 55-59 year old. These cohorts exhibited statistically significant negative NEAMR values across several seasons. Being born between 1986-1994 and 1961-1969, respectively, we cannot think of any concrete environmental causes for the consistent UM. One might speculate that either demographic resilience, lower exposure to iatrogenic or infectious risks, or differential mRNA-I timing or uptake played a role. Alternatively, selection bias in survival (e.g., the healthiest individuals being over-represented in these cohorts due to prior low mortality) could also contribute. Yet, both of these hypotheses are purely based on plausibility checks, which ought to be investigated more thoroughly.

Taken together, the NEAMR patterns observed over a quarter century suggest that EAM and UM are not randomly distributed across age groups and time, but instead reflect a complex interplay of cohort-specific vulnerabilities, pandemic policy exposures, and biological stressors. While this analysis cannot establish causality, it explores potential associations and supports the formulation of hypotheses to be tested in future work. In particular, we encourage the investigation of long-term health trajectories of historically distinct birth cohorts – especially in light of the post-2021 mortality anomalies among middle-aged adults and the previously war-affected elderly.

## 4.3 Cross-correlational evidence linking mortality to incidence and injection timelines

To evaluate possible associations between EAM and epidemic markers, we analysed the correlation coefficients between NEAMR time courses and those of PCR-incidence rates, mRNA-I incidences, and their combinations. In accordance with our methodological framework (Sect 2.5), we restrict our focus to negative time lags in the cross-correlation functions, which represent the temporal precedence of potential causative events (PCR-incidence or mRNA injections) over mortality responses (Figs 7 and 8).

Two intuitive expectations guided this analysis: first, that PCR-incidences should correlate *positively* with NEAMR, since they mark the spread of infection; and second, that mRNA-I incidences should correlate *negatively*, as mRNA-I were designed to reduce infection severity and mortality.

Surprisingly, in the time interval CW04-CW42 in 2021 (Fig 7), i.e. during 'alpha' existence, the PCR-incidence of the 'alpha' variant (PCR+(alpha)) correlated only moderately positively (coefficients maximally 0.68) with NEAMR in the cohorts of the 40-49, 60-69, and 70-79 year old. In the youngest cohort (0-29), there was virtually no such correlation, while the 30-39 year old stood out with a notable positive coefficient of 0.81, peaking at a lag of -21 weeks. Evidently, PCR+(alpha) did not coincide with almost immediately subsequent EM, but correlated with such about five months later. Moreover, in this cohort and during this time interval, the early presence of the 'delta' variant (its incidence: PCR+(delta)) showed a correlation with NEAMR nearly as high (0.77 at -1 week). More notably, the correlation between NEAMR and the products of PCR+(alpha) incidence and rates of mRNA-I1 or mRNA-I2, respectively, was

even higher, namely, 0.84 and 0.86, respectively. All this strongly suggests a harmful synergistic effect: mRNA-I administered at the height of 'alpha' circulation may have temporarily impaired immune function in this cohort, particularly in males (Fig F.12) more than females (Fig F.11). Consequently, the subsequent 'delta' wave, which began circulating in June 2021, may have disproportionately affected individuals with recently altered or weakened immune responses.

This interpretation gains plausibility from several additional observations. First, the emergence of 'delta' was temporally aligned with the peak of mRNA-I1 and mRNA-I2 rates in the 0-29 cohort and mRNA-I2 rates in the 40-59 year old. Second, the first PCR+(delta) incidence peaks occurred earliest and strongest in the cohorts under 40 years old (Fig 6). These temporal dynamics are compatible with the hypothesis that the 'delta' variant arose, at least in part, as a mutational response to high-volume SP exposure via mass mRNA-I campaigns during an active 'alpha' epidemic – a scenario contradicting classical epidemiological theory of appropriate inoculation strategies [39–41].

Further supporting this concern, exceptionally high NEAMR levels were observed during the second 'delta' wave in late 2021 (in the time interval from CW30 in 2021 to CW03 in 2022; Fig 8), coinciding with comparably high PCR+(delta) incidence (Fig 6) and strong to very strong positive correlations with NEAMR (coefficients 0.9-0.99, with lags 0-1 week) in all cohorts above age 40. The 30-39 year old again showed a weaker, though still considerable, correlation (0.82 at lag -5 weeks), while the youngest cohort (0-29) remained essentially uncorrelated.

An unexpected inverse correlation was noted in the eldest cohort (95+) during the 'alpha' period (Fig 7, bottom panel): higher PCR-incidences were associated with lower NEAMR values. On closer inspection, this counterintuitive finding appears to be a statistical artefact. The peak in EAM among the eldest occurred during the 'second wave' in December 2020, followed by a characteristic (natural) dip to UM in early 2021 – a seasonal pattern well-documented for severe flu seasons. The period of sharply ceasing NEAMR coincided with the start of both 'alpha' circulation and the earliest stages of the mRNA-I campaign, which was prioritised for the elderly. Thus, this resulting (intuitively expectable) negative correlation with mRNA-I reflects a likely coincidental temporal overlap rather than a causal protective effect of mRNA-I.

In all other age cohorts, and across both time intervals studied, the correlations between NEAMR and mRNA-I rates were consistently positive – contrary to the expected protective effect. This pattern holds for both mRNA-I1 and mRNA-I2 during 2021, and for mRNA-I3 during the 'booster' campaign of late 2021 into 2022. Indeed, mRNA-I3 incidence was strongly positively correlated with NEAMR in all cohorts above age 50 (Fig 8).

This alignment of mRNA-I campaigns with subsequent AM surges – most notably, the unprecedented December 2022 EAM peak (Fig 6) – casts serious doubt on the population-level efficacy of the mRNA-I programme in reducing mortality. On the contrary, our findings support the concern that mass mRNA-I during active viral spread, particularly with novel, mRNA-based platforms and incomplete immunological understanding, may have produced unintended adverse outcomes, detectable in terms of significant EAM.

These observations are further corroborated by independent analyses. Kuhbandner and Reitzner [7] reported a positive correlation between regional mRNA-I coverage and changes in NEAMR across Germany's 16 federal states. A similar association was observed for still-birth rates [5]. While correlation does not imply causation, these robust, temporally and geographically consistent patterns warrant urgent further investigation.

## 5 Limitations

Several limitations of this study should be acknowledged.

First, the demographic and PCR incidence data required pre-processing steps, including spline interpolation to harmonise differing age-cohort resolutions. While this approach is common in time-series analysis – and not as prone to error as spline extrapolation – it remains a modeling choice that could affect the precise alignment of signals.

Second, our analyses assume an age-independent distribution of SARS-CoV-2 variant incidence. In reality, variant spread and susceptibility may vary across age cohorts, and incorporating age-specific variant data could refine future analyses.

Third, the mortality baseline is modeled under an exponential decay assumption. This functional form is widely used in actuarial and epidemiological models [5], yet it may oversimplify complex demographic processes.

Fourth, the use of Pearson correlation coefficients to explore associations between incidence metrics and EAM has inherent limitations. Correlation coefficients quantify linear association but do not control for confounding. Moreover, lagged correlations in time series may be inflated by auto-correlation or shared trends, and negative correlations can arise as statistical artifacts.

Fifth, uncertainty exists regarding the timing of mRNA-I relative to mortality outcomes, particularly given delays between injection, reporting, and death registration. While our alignment strategy captures broad temporal patterns, the precise timing of individual-level events cannot be resolved in aggregate data.

Finally, regarding the observed temporal alignments between the mRNA-I campaign and mortality surges, the present study provides an important hypothesis-generating foundation rather than a closed conclusion. Yet, given the strength and consistency of the correlations across age cohorts and time windows, together with plausible biological mechanisms proposed in the literature, these findings form a hypothesis that warrants systematic investigation.

## 6 Conclusion

Our presented methodology reveals patterns of statistically significant excess all-cause and under-mortality across distinct age cohorts in Germany from 2000 through 2024. Notably, from autumn 2021 onward, adults aged 35-49 – representing the majority of school-aged children's parents – have experienced persistent and recurrent EAM. This signal is historically unprecedented in that particular age group and concealed in whole-population aggregates due to relatively low absolute mortality.

These findings call for urgent and transparent scientific investigation. In light of the temporal proximity between these mortality anomalies and the nationwide rollout of SARS-CoV-2 mRNA injections – particularly during periods of ongoing viral transmission – possible immunological or iatrogenic contributions must be examined. The consistency, magnitude, and demographic specificity of the observed mortality patterns warrant further investigation as statistical noise or incidental variation seem unlikely. Rather, they demand careful scrutiny from health authorities and policymakers. Future research should prioritise granular, cohort-resolved analyses over aggregate national statistics to detect, understand and explain such associations.

## Supporting information

**S1 Data. The following data tables are provided in machine-readable format (CSV) in the online repository.** For each sex and 10-year age cohort, they contain cohort sizes, weekly

death counts, all-cause mortality rates (AMR) PCR-incidence, mRNA-I rates, and modelled excess mortality rates.

- `Data_all_age_0-29.csv`
- `Data_all_age_30-39.csv`
- `Data_all_age_40-49.csv`
- `Data_all_age_50-59.csv`
- `Data_all_age_60-69.csv`
- `Data_all_age_70-79.csv`
- `Data_all_age_80+.csv`
- `Data_all_population.csv`
- `Data_female_age_0-29.csv`
- `Data_female_age_30-39.csv`
- `Data_female_age_40-49.csv`
- `Data_female_age_50-59.csv`
- `Data_female_age_60-69.csv`
- `Data_female_age_70-79.csv`
- `Data_female_age_80+.csv`
- `Data_female_population.csv`
- `Data_male_age_0-29.csv`
- `Data_male_age_30-39.csv`
- `Data_male_age_40-49.csv`
- `Data_male_age_50-59.csv`
- `Data_male_age_60-69.csv`
- `Data_male_age_70-79.csv`
- `Data_male_age_80+.csv`
- `Data_male_population.csv`

(ZIP)

**S2 Fig. The following figures are provided in the supplementary PDF file. Fig A.1.** *Females*: German expected normalised AMRs (exNAMRs) with 5-year resolution of age cohorts; an exNAMR course is plotted over the years from 2000 through 2024 for a given CW (one line); see again the caption of Fig 1) for details on normalisation, fitting, and extrapolation of these courses. **Fig A.2.** *Males*: German expected normalised AMRs (exNAMRs) with 5-year resolution of age cohorts; an exNAMR course is plotted over the years from 2000 through 2024 for a given CW (one line); see again the caption of Fig 1) for details on normalisation, fitting, and extrapolation of these courses. **Fig B.3.** *Females*: German weekly observed AMCs (magenta lines), and correspondingly expected values (exAMCs, black lines) from 2000 through 2024, for the same seven age cohorts as in Figs 2, 3, 6. **Fig B.4.** *Males*: German weekly observed AMCs (magenta lines), and correspondingly expected values (exAMCs, black lines) from 2000 through 2024, for the same seven age cohorts as in Figs 2, 3, 6. **Fig C.5.** *Females*: German weekly NEAMRs (black lines) from 2000 through 2024, for the same seven age cohorts as in Figs 2, 3, 6; values exceeding (red) or dropping below (green) the 95% CI indicated by a spot. **Fig C.6.** *Males*: German weekly NEAMRs (black lines) from 2000 through 2024, for the same seven age cohorts as in Figs 2, 3, 6; values exceeding (red) or dropping below (green) the 95% CI indicated by a spot. **Fig D.7.** *Females*: German seasonal NEAMRs from 2000 through 2024, with 5-year resolution of age cohorts; NEAMR values calculated within three seasons constituting a year (**top**): 'flu season 1' (fls$_1$: CW04-CW20), 'summer season' (sus: CW21-CW39), and 'flu season 2' (fls$_2$: CW40-subsCW03); NEAMR values exceeding (reddish) or dropping below (greenish) the 95% CI (determined for each season) indicated by an asterisk or circle, respectively; **bottom**: the corresponding histograms for the time spans 2000-2019 (grey) and 2020-2024 (magenta), respectively, their arithmetic mean values symbolised by solid vertical lines; the significance of the difference in mean values is indicated by star symbols: $p < 0.05$ (one), $p < 0.01$ (two), $p < 0.001$ (three). **Fig D.8.** *Males*: German seasonal NEAMRs from 2000 through 2024, with 5-year resolution of age cohorts; NEAMR values calculated within three seasons constituting a year (**top**): 'flu season 1' (fls$_1$: CW04-CW20), 'summer season' (sus: CW21-CW39), and 'flu season 2' (fls$_2$: CW40-subsCW03); NEAMR values exceeding (reddish) or dropping below (greenish) the 95% CI (determined for each season) indicated by an asterisk or circle, respectively; **bottom**: the corresponding histograms for the time spans 2000-2019 (grey) and 2020-2024 (magenta), respectively, their arithmetic mean values symbolised by solid vertical lines; the significance of the difference in

mean values is indicated by star symbols: $p < 0.05$ (one), $p < 0.01$ (two), $p < 0.001$ (three). **Fig E.9.** *Females*: German weekly NEAMRs (black lines) from 2020 through 2024, for the same seven age cohorts as in Figs 2, 3, 6; see caption of Fig 6 for detailed explanation of all symbols shown. **Fig E.10.** *Males*: German weekly NEAMRs (black lines) from 2020 through 2024, for the same seven age cohorts as in Figs 2, 3, 6; see caption of Fig 6 for detailed explanation of all symbols shown. **Fig F.11.** In Germany, during the time interval CW04-CW42,2021, and for seven *female* age cohorts, the (Pearson) coefficients of cross-correlating their respective weekly NEAMR time course with several (time-lagged) normalised incidence signals (regarding normalisation, see caption of Fig 6) are plotted: with the *normalised weekly numbers of positive PCR tests* PCR+(V) of two SARS-CoV-2 variants V∈{'alpha', 'delta'}, and with the *normalised weekly number of mRNA-I* of the first (mRNA-I1), and second (mRNA-I2) mRNA-I, as well as with two products by week of incidences; maximum and minimum coefficient values to the right of a sub-panel; the same for both sexes together: see again Fig 7. **Fig F.12.** In Germany, during the time interval CW04-CW42,2021, and for seven *male* age cohorts, the (Pearson) coefficients of cross-correlating their respective weekly NEAMR time course with several (time-lagged) normalised incidence signals (regarding normalisation, see caption of Fig 6) are plotted: with the *normalised weekly numbers of positive PCR tests* PCR+(V) of two SARS-CoV-2 variants V∈{'alpha', 'delta'}, and with the *normalised weekly number of mRNA-I* of the first (mRNA-I1), and second (mRNA-I2) mRNA-I, as well as with two products by week of incidences; maximum and minimum coefficient values to the right of a sub-panel; the same for both sexes together: see again Fig 7. **Fig G.13.** In Germany, during the time interval CW30,2021-CW03,2022, and for seven *female* age cohorts, the (Pearson) coefficients of cross-correlating their respective weekly NEAMR time course with several (time-lagged) normalised incidence signals (regarding normalisation, see caption of Fig 6) are plotted: with the *normalised weekly numbers of positive PCR tests* PCR+(V) of the SARS-CoV-2 variant V='delta', and with the *normalised weekly number of mRNA-I* of the first (mRNA-I1), second (mRNA-I2), and third (mRNA-I3) mRNA-I, as well as with one product by week of incidences; maximum and minimum coefficient values to the right of a sub-panel; the same for both sexes together: see again Fig 8. **Fig G.14.** In Germany, during the time interval CW30,2021-CW03,2022, and for seven *male* age cohorts, the (Pearson) coefficients of cross-correlating their respective weekly NEAMR time course with several (time-lagged) normalised incidence signals (regarding normalisation, see caption of Fig 6) are plotted: with the *normalised weekly numbers of positive PCR tests* PCR+(V) of the SARS-CoV-2 variant V='delta', and with the *normalised weekly number of mRNA-I* of the first (mRNA-I1), second (mRNA-I2), and third (mRNA-I3) mRNA-I, as well as with one product by week of incidences; maximum and minimum coefficient values to the right of a sub-panel; the same for both sexes together: see again Fig 8.
(PDF)

## Acknowledgements

We deeply thank a team of unknown authors at MWGFD e.V. for their sapient review [42] of a recent comprehensive microbiological analysis [22] on epigenetic modifications by the SARS-CoV-2 SP, and Harald Walach [43] for his sophisticated and embedding comment on that study and the MWGFD review. Also, our thanks go to Peter F. Meyer for both his utterly helpful comment [44] on [22], and, together with Sabine Stebel, their excellent elucidation [45] on further effects of the SP on immune responses, in particular, the production of IgG4 and their interaction with IgA antibodies.

## Author contributions

**Conceptualization:** Robert Rockenfeller, Michael Günther.

**Data curation:** Robert Rockenfeller.

**Formal analysis:** Robert Rockenfeller.

**Investigation:** Robert Rockenfeller.

**Methodology:** Robert Rockenfeller, Michael Günther.

**Software:** Robert Rockenfeller.

**Supervision:** Michael Günther.

**Validation:** Robert Rockenfeller.

**Visualization:** Robert Rockenfeller.

**Writing – original draft:** Robert Rockenfeller, Michael Günther.

**Writing – review & editing:** Robert Rockenfeller, Michael Günther.

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
