## [Decision Letter · Decision Letter 0]

15 Sep 2025

PONE-D-25-42536Cohort-resolved excess mortality in Germany (2000-2024): Patterns and implications for the SARS-CoV-2 eraPLOS ONE

Dear Dr. Rockenfeller,

Thank you for submitting your manuscript to PLOS ONE. After careful consideration, we feel that it has merit but does not fully meet PLOS ONE’s publication criteria as it currently stands. Therefore, we invite you to submit a revised version of the manuscript that addresses the points raised during the review process.

We look forward to receiving your revised manuscript.

Kind regards,

Kin Israel Notarte, MD

Guest Editor

PLOS ONE

2. Please include a separate caption for each figure in your manuscript.

Additional Editor Comments:

Reviewer #1:

Abstract

Lines 1–6: The abstract clearly explains the aim, method, and findings. However, the sentence on “associations between NEAMR and mRNA injection rates diverge from expected protective patterns” is strong and might suggest causation. Please soften this statement to avoid overstating conclusions.

Introduction

Lines 7–21: Clear background on how pandemics were defined. Sentences are long; consider splitting for easier reading.

Lines 22–29: Describes German government response. Important context, but the focus is political. Suggest shortening to keep attention on mortality data.

Lines 30–41: Useful explanation of excess mortality as a benchmark. However, very technical phrasing should be simplified for clarity.

Lines 42–50: Describes the 2021/22 season mortality. This is well explained but ends with speculation that “cannot be readily explained by viral characteristics alone.” This feels speculative; rephrase more cautiously.

Lines 51–64: Study objectives are clear. Good rationale for age-specific analysis.

Methods

Lines 66–80 (Abbreviations & Data Sources): Very detailed, but abbreviation-heavy. Consider moving some definitions to a supplementary table to improve readability.

Lines 81–101 (Demographic and PCR data): Clear description of data handling. But the spline interpolation (lines 95–98) is a modeling choice that could affect results; please justify this more.

Lines 102–111 (Variants): Assumes age-independent distribution. Note this as a limitation in the Discussion.

Lines 112–140 (Vaccination data): Very technical. Explains the grouping of age cohorts. The process is transparent, but long sentences make it hard to follow; consider simplifying.

Lines 141–169 (Mortality modeling): Clear description of the exponential fitting method. However, assuming mortality always follows an exponential decay may oversimplify. Please discuss this limitation.

Lines 183–214 (Cross-correlation analysis): Important part of the study. However, correlation does not equal causation. Please tone down interpretation and highlight confounding factors in the discussion.

Results

Lines 233–249 (Weekly expNAMRs): Technical but well described. Figures are necessary here.

Lines 482–508 (Infections & vaccination): Clear description of timing. Suggest highlighting uncertainty in vaccine timing vs. mortality patterns.

Lines 509–565 (Cross-correlations): Detailed. However, presenting strong positive correlations risks misinterpretation. Stress that correlation ≠ causation, and clarify that these are exploratory, not proof of effect.

Discussion

Lines 613–634: The authors suggest that repeated mRNA injections may cause immune fatigue and immune shifts (IgA/IgG4). This is speculative and should be clearly stated as a hypothesis, not a proven mechanism.

Lines 635–646: Good description of persistent excess mortality (EAM) in adults aged 35–49 from 2021–2024. This is a strong finding, well supported by the data.

Lines 647–657: Authors propose psychosocial stress (school closures, family stress) as a contributing factor. This is reasonable but should be supported with references to epidemiological studies.

Lines 695–699: Mentions under-mortality (UM) without clear cause. This is honest and appropriate, but the speculation (about resilience or survival bias) should be marked as uncertain.

Lines 707–733: The cross-correlation section interprets vaccine timing as possibly impairing immunity. This language is too strong—correlation does not prove effect. Please reframe as “possible associations that need further study.”

Lines 754–762: The authors themselves admit some negative correlations are statistical artifacts. This is good transparency and should be emphasized more to balance the stronger claims.

Lines 763–774: Concludes that vaccine campaigns align with mortality surges. This should be stated cautiously, as alignment in timing is not proof of cause.

Conclusion

Lines 790–793: Authors suggest a possible link to mRNA injections. This is speculative and goes beyond what the data can prove. Please tone this down and present it as an open question for further study, not a conclusion.

Lines 794–796: The wording “not easily dismissed” is too strong for a correlational study. Suggest softening to “warrants further investigation.”

Lines 797–799: The call for more granular, cohort-based research is excellent and should be the final emphasis, rather than speculation about causation.

Reviewer #2:

This study analysed all-cause mortality in Germany from 2000 to 2024 using data from a weekly, cohort-resolved framework across 15 age groups. The authors aimed to identify anomalies in specific age groups. To do so, they employed exponential trends derived from two decades of pre-pandemic data to spot patterns of excess and under-mortality before, during and after COVID-19 pandemic.

The authors reported sustained excess mortality among adults aged 35–49 and 75–79 from late 2021 through to 2024. Meanwhile, cohorts aged 30–34 and 55–59 exhibited persistent under-mortality.

Overall, if the research questions examined in this study were to be replicated in numerous robust studies, the findings could be significant for the implementation of public health interventions during pandemics such as COVID-19.

With that in mind, this reviewer has the following to remark:

1. Author summary

There is a risk that the “Author summary” section might seem redundant. The abstract serves as a concise summary of the study, and any important points mentioned in the “Author summary” can easily be incorporated into the different sections of the article.

2. Introduction

Towards the end of the introduction section, the authors write, “This study extends our previous work [4] by examining age-stratified, weekly EAM data in Germany from 2020 through 2024. Our goal is twofold. First, we investigate whether pandemic-level mortality signals may have been masked at the population level but remain detectable within specific age cohorts. Second, we assess whether age-resolved EAM correlates temporally with other epidemiological variables, in particular, PCR-confirmed SARS-CoV-2 incidence [7], and mRNA-I rates during the global immunisation campaign.”

The authors here clearly state their aim for conducting this study, which is appropriate.

However, before presenting their methods, they add this at the end of the introduction section:

“Our primary tool is the robust and minimally assumption-laden indicator of EAM, analysed at weekly resolution. We report not only significant seasonal EAM in older cohorts (as typically seen with influenza), but also unusually persistent and significant EAM among individuals aged 35 to 49 years – particularly from autumn 2021 onward, with distinct waves extending into 2024. Cross-correlation analyses suggest that these EAM patterns may be temporally associated with infection and mRNA-I waves, raising questions that warrant thorough further investigation.”

Thus far, the authors have referenced their results in the abstract, author summary, and introduction. It should be noted that no results from this study are allowed in the introduction section. The primary objective of this section is to articulate the authors’ intentions explicitly and end there.

3. Methods

3.1

In subsection 2.2.1, the authors state, “Earlier (2000-2020) weekly AMC data were taken from the Destatis (German o�ce for national statistics) webpage at [8]. More current (2021-2024) weekly AMC data were likewise taken from the Destatis webpage at [9].”

It is not necessary to include “at” followed by the citation. The citation alone, after the webpage, is sufficient.

3.2

In subsection 2.2.2, the same issue as above is observed:

“The demographic (age cohort) distribution between 2000 and 2020 (based on the 2011 census) and its presumable distribution onwards (based on the most plausible scenario, the default model variant V1) were taken from the Destatis webpage at [10].”

3.3

In subsection 2.3, the authors state, “The value of bcoh can be interpreted as a asymptotical limit for large t, whereas…”

It should be “an asymptotic limit,” or if the authors prefer “an asymptotical limit,” not “a asymptotical limit.”

3.4

In subsection 2.5, the authors state, “Interpretation of the cross-correlation follows standard convention: A correlation coe�cient of 0 indicates no linear statistical association.”

Do the authors mean that a correlation coefficient not equal to zero indicates a linear relationship? Do Figures 7 and 8 support such assumptions and their choice of correlation analysis method?

It is well-known that using a lag-dependent correlation structure can lead to biased estimates due to violations of the classical linear model assumptions. Another challenge is model complexity: the more lags, the more complex the model becomes, making the results more difficult to interpret. Consequently, it is important to consider what measures the authors took to avoid these issues.

4. Results

This section needs editing to make the text more concise and improve readability.

5. Discussion

5.1

In the discussion section, the authors state, “This cohort was born in 1941-1949, i.e. during World War II and directly after, and would have partly lived their early child and formative years in post-war Europe, a time of reconstruction, nutritional scarcity, and psychological upheaval.”

This sentence requires attention. It could be edited to read, “This cohort was born between 1941 and 1949, during and immediately after World War II. They would have spent part of their early childhood and formative years in post-war Europe — a time of reconstruction, nutritional scarcity and psychological upheaval.”

5.2

In subsection 4.3, the authors note, “To evaluate possible associations between EAM and epidemic markers, we analysed the correlation coe�cients between NEAMR time courses and those of PCR-positive (PCR+) test rates, mRNA-I incidences, and their combinations. In accordance with our methodological framework (Sec. 2.5), we restrict our focus to negative time lags in the cross-correlation functions, which represent the temporal precedence of potential causative events (infections or injections) over mortality responses (Figs. 7, 8).”

In addition, in subsection 2.5, the authors had already stated, “This reflects a causal hypothesis framework: that infection or injection events may precede and potentially influence AM trends, whereas reverse causation – NEAMR causing PCR or mRNA-I incidence – is implausible at the population level.”

Furthermore, in subsection 3.6, the authors underline, “We focus exclusively on negative time lags, which reflect the potential causal influence of these factors on subsequent mortality...”

It is important to note that the above statements are controversial, and the phrasing are a surprisingly common pitfall in correlation studies.

As has been noted many times before, correlations are not causations. Simply because two variables are highly correlated doesn’t mean one causes the other. Even with a very high Pearson coefficient, the apparent correlation can be purely coincidental, influenced by other hidden variables—a phenomenon known as spurious correlations.

In general, caution should be exercised when making claims about causality. Therefore, even though the authors acknowledge that “correlation does not imply causation” at the end of their discussion section, every sentence about “causative,” “causal” and “causation” in this paper needs to be rephrased for clarity.

5.3

Furthermore, it is noteworthy that this paper does not address any limitations of the study conducted. For example, the authors could have included the limitations of using Pearson correlation coefficients to assess a potential association between cohort-level AM patterns and PCR-confirmed SARS-CoV-2 infection or mRNA-I activity.

If the authors deliberately omitted any limitations from their paper, the reason for this remains unclear.

Finally, I would like to commend the authors for their call for urgent, transparent scientific investigations into this topic. I hope my review is helpful and wish the authors the very best with their research!

Reviewers' comments:

Reviewer's Responses to Questions

**Comments to the Author**

1. Is the manuscript technically sound, and do the data support the conclusions?

Reviewer #1: No

Reviewer #2: Yes

2. Has the statistical analysis been performed appropriately and rigorously? 

Reviewer #1: No

Reviewer #2: Yes

3. Have the authors made all data underlying the findings in their manuscript fully available?

Reviewer #1: Yes

Reviewer #2: Yes

4. Is the manuscript presented in an intelligible fashion and written in standard English?

Reviewer #1: Yes

Reviewer #2: Yes

5. Review Comments to the Author

Reviewer #1: Abstract

Lines 1–6: The abstract clearly explains the aim, method, and findings. However, the sentence on “associations between NEAMR and mRNA injection rates diverge from expected protective patterns” is strong and might suggest causation. Please soften this statement to avoid overstating conclusions.

Introduction

Lines 7–21: Clear background on how pandemics were defined. Sentences are long; consider splitting for easier reading.

Lines 22–29: Describes German government response. Important context, but the focus is political. Suggest shortening to keep attention on mortality data.

Lines 30–41: Useful explanation of excess mortality as a benchmark. However, very technical phrasing should be simplified for clarity.

Lines 42–50: Describes the 2021/22 season mortality. This is well explained but ends with speculation that “cannot be readily explained by viral characteristics alone.” This feels speculative; rephrase more cautiously.

Lines 51–64: Study objectives are clear. Good rationale for age-specific analysis.

Methods

Lines 66–80 (Abbreviations & Data Sources): Very detailed, but abbreviation-heavy. Consider moving some definitions to a supplementary table to improve readability.

Lines 81–101 (Demographic and PCR data): Clear description of data handling. But the spline interpolation (lines 95–98) is a modeling choice that could affect results; please justify this more.

Lines 102–111 (Variants): Assumes age-independent distribution. Note this as a limitation in the Discussion.

Lines 112–140 (Vaccination data): Very technical. Explains the grouping of age cohorts. The process is transparent, but long sentences make it hard to follow; consider simplifying.

Lines 141–169 (Mortality modeling): Clear description of the exponential fitting method. However, assuming mortality always follows an exponential decay may oversimplify. Please discuss this limitation.

Lines 183–214 (Cross-correlation analysis): Important part of the study. However, correlation does not equal causation. Please tone down interpretation and highlight confounding factors in the discussion.

Results

Lines 233–249 (Weekly expNAMRs): Technical but well described. Figures are necessary here.

Lines 482–508 (Infections & vaccination): Clear description of timing. Suggest highlighting uncertainty in vaccine timing vs. mortality patterns.

Lines 509–565 (Cross-correlations): Detailed. However, presenting strong positive correlations risks misinterpretation. Stress that correlation ≠ causation, and clarify that these are exploratory, not proof of effect.

Discussion

Lines 613–634: The authors suggest that repeated mRNA injections may cause immune fatigue and immune shifts (IgA/IgG4). This is speculative and should be clearly stated as a hypothesis, not a proven mechanism.

Lines 635–646: Good description of persistent excess mortality (EAM) in adults aged 35–49 from 2021–2024. This is a strong finding, well supported by the data.

Lines 647–657: Authors propose psychosocial stress (school closures, family stress) as a contributing factor. This is reasonable but should be supported with references to epidemiological studies.

Lines 695–699: Mentions under-mortality (UM) without clear cause. This is honest and appropriate, but the speculation (about resilience or survival bias) should be marked as uncertain.

Lines 707–733: The cross-correlation section interprets vaccine timing as possibly impairing immunity. This language is too strong—correlation does not prove effect. Please reframe as “possible associations that need further study.”

Lines 754–762: The authors themselves admit some negative correlations are statistical artifacts. This is good transparency and should be emphasized more to balance the stronger claims.

Lines 763–774: Concludes that vaccine campaigns align with mortality surges. This should be stated cautiously, as alignment in timing is not proof of cause.

Conclusion

Lines 790–793: Authors suggest a possible link to mRNA injections. This is speculative and goes beyond what the data can prove. Please tone this down and present it as an open question for further study, not a conclusion.

Lines 794–796: The wording “not easily dismissed” is too strong for a correlational study. Suggest softening to “warrants further investigation.”

Lines 797–799: The call for more granular, cohort-based research is excellent and should be the final emphasis, rather than speculation about causation.

Reviewer #2: This study analysed all-cause mortality in Germany from 2000 to 2024 using data from a weekly, cohort-resolved framework across 15 age groups. The authors aimed to identify anomalies in specific age groups. To do so, they employed exponential trends derived from two decades of pre-pandemic data to spot patterns of excess and under-mortality before, during and after COVID-19 pandemic.

The authors reported sustained excess mortality among adults aged 35–49 and 75–79 from late 2021 through to 2024. Meanwhile, cohorts aged 30–34 and 55–59 exhibited persistent under-mortality.

Overall, if the research questions examined in this study were to be replicated in numerous robust studies, the findings could be significant for the implementation of public health interventions during pandemics such as COVID-19.

With that in mind, this reviewer has the following to remark:

1. Author summary

There is a risk that the “Author summary” section might seem redundant. The abstract serves as a concise summary of the study, and any important points mentioned in the “Author summary” can easily be incorporated into the different sections of the article.

2. Introduction

Towards the end of the introduction section, the authors write, “This study extends our previous work [4] by examining age-stratified, weekly EAM data in Germany from 2020 through 2024. Our goal is twofold. First, we investigate whether pandemic-level mortality signals may have been masked at the population level but remain detectable within specific age cohorts. Second, we assess whether age-resolved EAM correlates temporally with other epidemiological variables, in particular, PCR-confirmed SARS-CoV-2 incidence [7], and mRNA-I rates during the global immunisation campaign.”

The authors here clearly state their aim for conducting this study, which is appropriate.

However, before presenting their methods, they add this at the end of the introduction section:

“Our primary tool is the robust and minimally assumption-laden indicator of EAM, analysed at weekly resolution. We report not only significant seasonal EAM in older cohorts (as typically seen with influenza), but also unusually persistent and significant EAM among individuals aged 35 to 49 years – particularly from autumn 2021 onward, with distinct waves extending into 2024. Cross-correlation analyses suggest that these EAM patterns may be temporally associated with infection and mRNA-I waves, raising questions that warrant thorough further investigation.”

Thus far, the authors have referenced their results in the abstract, author summary, and introduction. It should be noted that no results from this study are allowed in the introduction section. The primary objective of this section is to articulate the authors’ intentions explicitly and end there.

3. Methods

3.1

In subsection 2.2.1, the authors state, “Earlier (2000-2020) weekly AMC data were taken from the Destatis (German o�ce for national statistics) webpage at [8]. More current (2021-2024) weekly AMC data were likewise taken from the Destatis webpage at [9].”

It is not necessary to include “at” followed by the citation. The citation alone, after the webpage, is sufficient.

3.2

In subsection 2.2.2, the same issue as above is observed:

“The demographic (age cohort) distribution between 2000 and 2020 (based on the 2011 census) and its presumable distribution onwards (based on the most plausible scenario, the default model variant V1) were taken from the Destatis webpage at [10].”

3.3

In subsection 2.3, the authors state, “The value of bcoh can be interpreted as a asymptotical limit for large t, whereas…”

It should be “an asymptotic limit,” or if the authors prefer “an asymptotical limit,” not “a asymptotical limit.”

3.4

In subsection 2.5, the authors state, “Interpretation of the cross-correlation follows standard convention: A correlation coe�cient of 0 indicates no linear statistical association.”

Do the authors mean that a correlation coefficient not equal to zero indicates a linear relationship? Do Figures 7 and 8 support such assumptions and their choice of correlation analysis method?

It is well-known that using a lag-dependent correlation structure can lead to biased estimates due to violations of the classical linear model assumptions. Another challenge is model complexity: the more lags, the more complex the model becomes, making the results more difficult to interpret. Consequently, it is important to consider what measures the authors took to avoid these issues.

4. Results

This section needs editing to make the text more concise and improve readability.

5. Discussion

5.1

In the discussion section, the authors state, “This cohort was born in 1941-1949, i.e. during World War II and directly after, and would have partly lived their early child and formative years in post-war Europe, a time of reconstruction, nutritional scarcity, and psychological upheaval.”

This sentence requires attention. It could be edited to read, “This cohort was born between 1941 and 1949, during and immediately after World War II. They would have spent part of their early childhood and formative years in post-war Europe — a time of reconstruction, nutritional scarcity and psychological upheaval.”

5.2

In subsection 4.3, the authors note, “To evaluate possible associations between EAM and epidemic markers, we analysed the correlation coe�cients between NEAMR time courses and those of PCR-positive (PCR+) test rates, mRNA-I incidences, and their combinations. In accordance with our methodological framework (Sec. 2.5), we restrict our focus to negative time lags in the cross-correlation functions, which represent the temporal precedence of potential causative events (infections or injections) over mortality responses (Figs. 7, 8).”

In addition, in subsection 2.5, the authors had already stated, “This reflects a causal hypothesis framework: that infection or injection events may precede and potentially influence AM trends, whereas reverse causation – NEAMR causing PCR or mRNA-I incidence – is implausible at the population level.”

Furthermore, in subsection 3.6, the authors underline, “We focus exclusively on negative time lags, which reflect the potential causal influence of these factors on subsequent mortality...”

It is important to note that the above statements are controversial, and the phrasing are a surprisingly common pitfall in correlation studies.

As has been noted many times before, correlations are not causations. Simply because two variables are highly correlated doesn’t mean one causes the other. Even with a very high Pearson coefficient, the apparent correlation can be purely coincidental, influenced by other hidden variables—a phenomenon known as spurious correlations.

In general, caution should be exercised when making claims about causality. Therefore, even though the authors acknowledge that “correlation does not imply causation” at the end of their discussion section, every sentence about “causative,” “causal” and “causation” in this paper needs to be rephrased for clarity.

5.3

Furthermore, it is noteworthy that this paper does not address any limitations of the study conducted. For example, the authors could have included the limitations of using Pearson correlation coefficients to assess a potential association between cohort-level AM patterns and PCR-confirmed SARS-CoV-2 infection or mRNA-I activity.

If the authors deliberately omitted any limitations from their paper, the reason for this remains unclear.

Finally, I would like to commend the authors for their call for urgent, transparent scientific investigations into this topic. I hope my review is helpful and wish the authors the very best with their research!

6. PLOS authors have the option to publish the peer review history of their article (what does this mean?). If published, this will include your full peer review and any attached files.

Reviewer #1: No

Reviewer #2: **Yes: **Dr. Widad Akreyi

---

## [Author Response · Author response to Decision Letter 1]

24 Sep 2025

All responses to the reviewers' comments can be found in the corresponding pdf.

A tracked-change as well as a cleaned version of the manuscript are available.

We sincerely thank the reviewers for their time and constructive effort.

---

## [Decision Letter · Decision Letter 1]

6 Oct 2025

Cohort-resolved excess mortality in Germany (2000-2024): Patterns and implications for the SARS-CoV-2 era

PONE-D-25-42536R1

Dear Dr. Rockenfeller,

We’re pleased to inform you that your manuscript has been judged scientifically suitable for publication and will be formally accepted for publication once it meets all outstanding technical requirements.

Kind regards,

Kin Israel Notarte, MD

Guest Editor

PLOS ONE

Additional Editor Comments (optional):

Reviewers' comments:

Reviewer's Responses to Questions

**Comments to the Author**

1. If the authors have adequately addressed your comments raised in a previous round of review and you feel that this manuscript is now acceptable for publication, you may indicate that here to bypass the “Comments to the Author” section, enter your conflict of interest statement in the “Confidential to Editor” section, and submit your "Accept" recommendation.

Reviewer #1: All comments have been addressed

Reviewer #2: (No Response)

2. Is the manuscript technically sound, and do the data support the conclusions?

Reviewer #1: (No Response)

Reviewer #2: Yes

3. Has the statistical analysis been performed appropriately and rigorously? 

Reviewer #1: (No Response)

Reviewer #2: Yes

4. Have the authors made all data underlying the findings in their manuscript fully available?

Reviewer #1: (No Response)

Reviewer #2: Yes

5. Is the manuscript presented in an intelligible fashion and written in standard English?

Reviewer #1: (No Response)

Reviewer #2: Yes

6. Review Comments to the Author

Reviewer #1: (No Response)

Reviewer #2: Following recommendations from the editor and reviewers, the authors have made modifications to their paper that have improved it.

I wish the authors the very best with their research!

7. PLOS authors have the option to publish the peer review history of their article (what does this mean?). If published, this will include your full peer review and any attached files.

Reviewer #1: No

Reviewer #2: **Yes: **Dr. Widad Akreyi

---

## [Editor Report · Acceptance letter]

PONE-D-25-42536R1

PLOS ONE

Dear Dr. Rockenfeller,

I'm pleased to inform you that your manuscript has been deemed suitable for publication in PLOS ONE. Congratulations! Your manuscript is now being handed over to our production team.

Kind regards,

on behalf of

Dr. Kin Israel Notarte

Guest Editor

PLOS ONE